# Surface-Aware Feed-Forward Quadratic Gaussian for Frame Interpolation with Large Motion

Zaoming Yan [12]     Yaomin Huang[12]     Pengcheng Lei [12]     Qizhou Chen [1]

Guixu Zhang[12]

Faming Fang[12*]

[1] School of Computer Science and Technology, East China Normal University, Shanghai, China.

[2] The KLATASDS & Shanghai Key Laboratory of MIP.

{yan_zaoming,pengchenglei1995}@163.com   {ymhuang, qizhouchen}@stu.ecnu.edu.cn

{gxzhang, fmfang}@cs.ecnu.edu.cn

## Abstract

Large motion poses a critical challenge in Video Frame Interpolation (VFI) task, as it requires accurate modeling of object correspondences across frames. Existing methods primarily rely on convolutional or attention-based models, which operate at the pixel or patch level. This inherently limits them to local object correspondences, making it difficult to capture frame-level object correspondences and often leading to failure under large motion. Inspired by the fundamental theorem of surface, we explore frame-level object correspondences through the lens of differential surface. The core idea is to represent video frames as 3D surfaces and align them by matching their surface properties, thereby achieving global surface alignment and frame-level object alignment. To implement the core idea, we propose the Surface-Aware Feed-Forward Quadratic Gaussian framework, mainly consisting of the Feed-Forward Quadratic Gaussian and Surface Properties modules. Feed-Forward Quadratic Gaussian is designed to map frames to Quadratic Gaussian, which flexibly fits the object surface. Unlike previous methods that compute local correspondences, Surface Properties facilitates global surface-level alignment, which drives object correspondence alignment. Finally, we rasterize the surface properties onto the interpolated camera plane and define loss functions to supervise alignment explicitly. The outstanding performance on the large motion benchmark demonstrates the effectiveness of our framework.

## 1   Introduction

Video Frame Interpolation (VFI) is a fundamental low-level vision task that aims to increase the frame rate of a video by synthesizing intermediate frames between consecutive inputs. It has a wide range of real-world applications, including slow-motion video generation [1, 2, 3], video compression [4, 5], and novel view synthesis [6, 7, 8]. Despite recent progress, VFI remains challenging, particularly in large and complex motion commonly found in casually captured videos. More recently, the emergence of film agent frameworks [9, 10, 11] has introduced intelligent agents for cinematic content creation. In such scenarios, handling large motion is critical for tasks such as scene composition, camera planning, and visual continuity. These demands highlight the need for more robust VFI methods capable of modeling long-range object correspondences.

At its core, VFI requires establishing accurate correspondences between objects across frames [12]. Video frame interpolation methods can be broadly divided into two types: those based on

---

*Corresponding Author

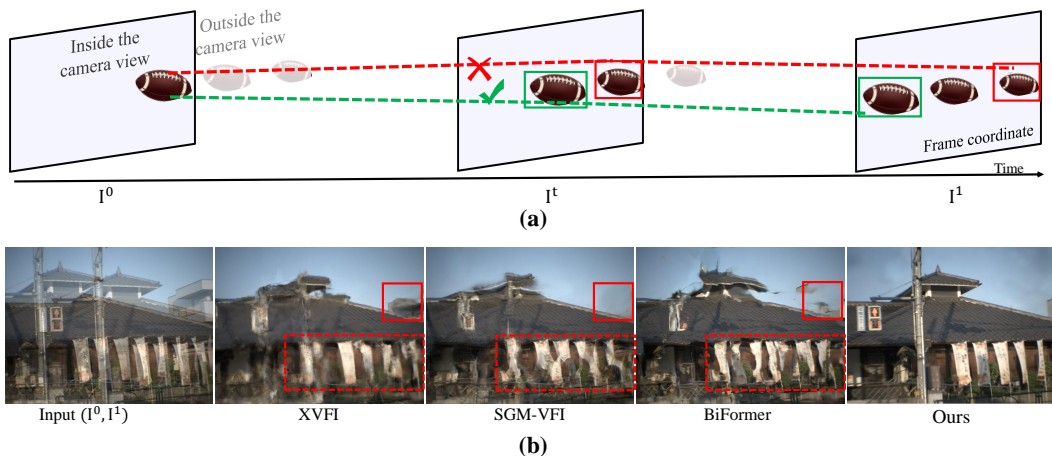

**(a)**

| Input ($I^0, I^1$) | XVFI | SGM-VFI | BiFormer | Ours |

**(b)**

Figure 1: Challenge: Existing methods lack global frame-level object correspondences, which results in suboptimal matching.

Convolutional Neural Networks (CNNs) and those based on attention mechanisms. CNN-based methods [3, 13, 14, 15, 16, 17, 18, 12] estimate optical flow to synthesize intermediate frames, where the optical flow serves as a proxy for object correspondences. However, due to the local receptive field of convolution operations, CNN-based methods often compute object correspondences only at the pixel level, which limits their ability to capture global relationships. To overcome the locality of CNNs, attention-based architectures have been introduced for video frame interpolation [19, 20]. These methods divide input frames into patches and compute attention between them. However, this patch-level modeling still lacks a true global understanding of object correspondences across frames.

Upon scrutinizing and experimenting on the released implementations of existing methods [17, 15, 18, 12], we observe that they suffer from performance deterioration in large motion. As illustrated in Figure 1, in the large motion, existing approaches fail to establish accurate object correspondences, leading to misalignment results. We attribute these failures to the inherent locality of convolution- and attention-based methods, which operate at the pixel or patch level and thus struggle to model frame-level object correspondences. We further identify two representative failure cases (Figure 1 (b)): **(i)** Repeated objects across frames confuse optical flow estimation, resulting in incorrect local correspondences. For example, the red dashed lines show mismatched flags, causing fragmentation in the interpolated frame. **(ii)** Some objects required in the target frame are missing in adjacent frames, leading to ambiguous correspondences [21], as shown in the upper-right region.

As discussed above, overcoming the limitations of local object correspondences, particularly in the presence of large motion, necessitates the establishment of frame-level object correspondences. However, how to theoretically formulate and practically implement such correspondences requires further exploration and discussion. Inspired by the fundamental theorem of surfaces[22], we propose to represent video frames as differential surfaces (Figure 2), enabling the exploration of geometric constraints and frame-level correspondences. The core idea of this work is to map video frames onto 3D surfaces and align the overlapping regions using surface properties. This approach drives global surface alignment and promotes consistent geometric structure across frames. However, implementing this idea presents two key challenges: **(1)** how to effectively represent surface, and **(2)** which surface properties are most suitable for alignment.

To address the surface representation challenge, we propose the **Feed-Forward Quadratic Gaussian**. The Quadratic Gaussian Splatting (QGS) [23] defines a Gaussian distribution on a quadratic paraboloid, allowing smooth transitions between convex and concave forms for flexible surface fitting. However, QGS assumes accurate camera poses, which limits its applicability in real-world scenarios where videos are casually captured. To overcome this, Feed-Forward Quadratic Gaussian efficiently transforms such video frames into 3D surface representations without relying on strict assumptions such as accurate camera poses or depth maps.

To facilitate surface alignment, we leverage two key surface properties: **normal and curvature**. Surface normals are widely used to ensure geometric continuity across frames [24]. However, normal alone may become unreliable and introduce errors within large motions (surfaces with rapid bends). This motivates the incorporation of higher-order geometric descriptors such as curvature, which can

capture fine-grained surface structures. Therefore, we jointly use surface normals and curvature as consistency constraints to ensure robust surface alignment across frames.

To explicitly supervise surface continuity, we rasterize the normal and curvature onto the camera plane and define loss functions. By leveraging these modules, we build a novel pipeline, named **Surface-Aware Feed-Forward Quadratic Gaussian**. Our method achieves state-of-the-art performance on large motion benchmarks [12]. Furthermore, we conduct extensive ablation studies to validate the contribution of each component.

The contributions of this work are summarized as: **(1)** Inspired by the Fundamental Theorem of Surfaces, we introduce a differential surface to present a video, to explore frame-level object correspondences and geometric constraints. **(2)** This paper introduces a Surface-Aware Feed-Forward Quadratic Gaussian framework that maps video frames into 3D surfaces, aiming to overcome the limited local correspondences. **(3)** Our pipeline illustrates state-of-the-art performance on the large motion benchmark.

## 2 Related Work

**Video Frame Interpolation.** Recently, advancements in deep learning have led to various methods for video frame interpolation. These methods can be broadly categorized into two main paradigms: reconstruction-based [25, 15, 14, 26, 27, 16, 28, 29, 30] and denoising diffusion probabilistic model (DDPM)-based [31, 32, 33, 34, 35]. (i) Initially, DVF [36] utilized a U-Net-like network to model two input frames and predicted the voxel flow for warping the two frames into the intermediate frame. To obtain the optical flow from the intermediate frame to the input frames, [15, 37] proposed distillation strategies to obtain the optical flow from the intermediate frame to the input frames. In large-scale motion scenarios, methods such as SGM-VFI [12], FILM[18], and XVFI [17] leverage enhanced global information in optical flow to establish accurate frame-to-frame correspondences for objects. These Kernel-based methods are implemented as separable convolutions [38], deformable convolutions [39, 40, 41]. (ii) Based on Denoising Diffusion Probabilistic Models (DDPM)[42], leverage generative techniques like DDPM to fill occlusions caused by motion. These DDPM-based approaches are implemented as score-based diffusion [43, 44, 45], motion-aware diffusion [46, 47], and Brownian bridge diffusion [48, 49]. However, most DDPM-based methods are significantly time-consuming and challenging for real-time inference.

**3D Gaussian Splatting.** In recent years, 3D Gaussian splatting has emerged as an active area of research in the field of 3D reconstruction. Various approaches have been proposed across different domains, broadly categorized into static scene Gaussian splatting and dynamic scene Gaussian splatting. Gaussian Splatting [50] enhances rendering quality in radiance fields. To further adapt to diverse reconstruction scenarios, [51, 52, 53] have been proposed, significantly improving the generalization capability of 3DGS-based reconstruction. To accommodate dynamic scenes, [54, 55, 56, 57, 58, 59] has been extended to handle such environments. However, the per-scene optimization of 3DGS requires densely captured images and sparse point cloud generated by SfM for initialization. Recent works [60, 61, 62, 63, 64, 65] have explored feed-forward models for sparse-view Gaussian reconstruction by capitalizing on large-scale datasets and scalable model architectures[66, 67, 68].

## 3 Preliminary

### 3.1 3D Gaussian Splatting

Kerbl et al. [69] proposed representing a scene using 3D Gaussian ellipsoids as primitives and rendering images using differentiable volume splatting. Associates a 3D Gaussian $i$ with a position $\mu_i$, covariance matrix $\Sigma_i$, opacity $o_i$ and spherical harmonics (SH) coefficients $h_i$. The final opacity of a 3D Gaussian at any spatial point $\mathbf{p} = (x, y, z)$ is:

$$\alpha_i = o_i \underbrace{\exp\left(-\frac{1}{2}(\mathbf{p} - \mu_i)^T \Sigma^{-1}(\mathbf{p} - \mu_i)\right)}_{\mathcal{G}}, \tag{1}$$

where the covariance matrix $\Sigma = RSS^T R^T$, and $\mathcal{G}$ is Gaussian distribution.

To render an image, 3D Gaussians are first projected to 2D image space via an approximation of the perspective transformation. Specifically, the projection of a 3D Gaussian is approximated as a 2D

Gaussian with center $\mu_i^{2D}$ and covariance $\Sigma_i^{2D}$. Center $\mu_i^{2D}$ and covariance $\Sigma_i^{2D}$ are computed as

$$\mu_i^{2D} = (K(W\mu_i)/(W\mu_i)_z), \quad \Sigma^{2D} = JW\Sigma_i W^T J^T, \tag{2}$$

where $W$ is a transformation from the world space to the camera space, and $J$ is a local affine transformation.

After sorting the Gaussians in depth order, the color at a pixel is obtained by volume rendering:

$$I(u,v) = \sum_{i=0}^{N-1} \mathbf{c}_i \alpha_i^{2D} \prod_{j=0}^{i-1} (1 - \alpha_j^{2D}). \tag{3}$$

Here, $\alpha_i^{2D}$ is a 2D version of Eq. (1), with $\mu_i, \Sigma_i, \mathbf{P}$ replaced bu $\mu_i^{2D}, \Sigma_i^{2D}, (u,v)$ (pixel coordinate). $\mathbf{c}_i$ is the RGB color after evaluating SH with the view direction.

### 3.2 Quadratic Gaussian

Zhang et al. [23] proposed representing a scene using a Quadratic Gaussian as a surface and rendering images using differentiable volume splatting. For convenience, the Quadratic Gaussian distribution is expressed in cylindrical coordinates, and the opacity of a Quadratic Gaussian at any spatial point $\mathbf{p} = (\theta, \rho, z(\theta, \rho))$ is:

$$\alpha_i = o_i \underbrace{\exp\left(-\frac{(\mu_i(\theta, \rho))^2}{2(\sigma_i(\theta))^2}\right)}_{\mathcal{G}}, \tag{4}$$

$$\sigma_i(\theta) = \frac{s_1 s_2}{\sqrt{(s_2 \cos\theta)^2 + (s_1 \sin\theta)^2}}, \quad \mu_i(\theta, \rho) = \int_0^\rho \sqrt{1 + (2at)^2}\, dt \tag{5}$$

where $\mu_i(\cdot)$ [23] is the mean of the Gaussian distribution on the surface, and $\sigma_i$ is the covariance of the Gaussian distribution on the surface. $\mathcal{G}$ is defined as the corresponding Gaussian function. $S = diag(s_1, s_2, s_3)$ denotes the scale of the Quadratic Gaussian. $a(\cdot)$ is related to the coefficient of $\theta$.

To render an image, the Quadratic Gaussian follows the same 3D Gaussian splatting way, which is projected to 2D image space via an approximation of the perspective transformation. After splatting and sorting the Gaussians in depth order, the color at a pixel $(u, v)$ is obtained by rendering [70]:

$$I(u,v) = \sum_{i=0}^{N-1} \mathbf{c}_i \alpha_i^{2D} \prod_{j=1}^{i-1} \left(1 - \alpha_j^{2D}\right) \tag{6}$$

where $N$ denotes the pixel numbers of the rendered image.

## 4 Method

### 4.1 Problem formulation

**Frame Interpolation.**

In the video frame interpolation task, it can be written as

$$I^t = \mathbf{F}(I^0, I^1), t \in (0, 1), \tag{7}$$

where $I^0$ and $I^1$ are input frames. 0 and 1 are two input views and $t$ is an interpolated view index between 0 and 1. To synthesize an intermediate frame $I^t$ where $t \in (0, 1)$, existing algorithms [3, 16, 12, 29, 30] typically extract object correspondences between two consecutive frames, facilitating object alignment.

**Frame Interpolation under Differential Surface.**

In large motion scenes, video frame interpolation methods often fall into suboptimal object correspondences, as they operate at the pixel level or patch level and thus struggle to model frame-level object correspondences.

To address the limitations of local object correspondences, we redefine the large motion problem as a global frame-level alignment task by aligning the surfaces that represent each frame. The core idea of frame-level alignment is that aligning the geometric properties between surfaces (such as normals and curvatures) can facilitate the alignment between surfaces, as illustrated in Figure 2.

In the following sections, we describe how to construct surface representations from video frames and how to select surface properties to facilitate accurate correspondences across surfaces.

Figure 2: The core idea is mapping video frames into 3D surfaces and aligning them by matching surface properties, leading to global surface-level alignment.

### 4.2 Surface-Aware Feed-Forward Quadratic Gaussian

Although modeling object correspondences through differential surface representations is theoretically reasonable, it remains challenging to implement: **1)** What type of 3D primitives is suitable for representing a differential surface? **2)** Which surface properties should be selected to facilitate alignment between differential surfaces?

#### 4.2.1 Feed-Forward Quadratic Gaussian

Modeling the texture and surface details in videos remains challenging for 3D primitives such as point clouds and 3DGS, which often struggle to accurately represent complex surfaces. Fortunately, Quadratic Gaussian Splatting (QGS) [23] is defined on a paraboloid and constructs Gaussian distributions based on geodesic distances. This enables the energy of the Gaussians to be concentrated on the surface, thereby effectively capturing complex surface and textural details. However, existing methods such as 3DGS [50], 2DGS [71], and QGS [23] still rely on accurate camera poses, which are difficult to obtain in sparse-view or unconstrained settings, thereby limiting their practical applicability. To overcome this limitation, the practical **Feed-Forward Quadratic Gaussian** is introduced that efficiently transforms frames into 3D surface representations.

Given a set of video frames, the goal of Feed-Forward Quadratic Gaussian is to generate the object surface in the QGS representation. Unlike prior methods, it does not require additional data such as camera poses, enabling single feed-forward inference. Feed-Forward Quadratic Gaussian mainly includes two sub-models: a backbone and a Quadratic Gaussian head. Formally, Feed-Forward Quadratic Gaussian can be written as:

$$f_\theta : \{I^0, I^1\} \mapsto \{\mathcal{P}^0, \mathcal{P}^1, \mathcal{C}^0, \mathcal{C}^1, \mathcal{F}^0, \mathcal{F}^1\}, \quad h_\theta : \{\mathcal{P}^0, \mathcal{P}^1, \mathcal{F}^0, \mathcal{F}^1\} \mapsto \{\mathcal{G}^0, \mathcal{G}^1\}, \quad (8)$$

$f_\theta$ is the backbone and $h_\theta$ is the Quadratic Gaussian Head. $\mathcal{F}^0, \mathcal{F}^1$ is the frame features. $\mathcal{P}^0, \mathcal{P}^1$ is the 3D point clouds. $\mathcal{C}^0, \mathcal{C}^1$ is the camera parameters. $\mathcal{G}^0, \mathcal{G}^1$ is the Quadratic Gaussian.

**Backbone.** Foundation models for 3D reconstruction (e.g., DUSt3R[66], MASt3R [67]) have shown remarkable competitiveness and superior performance in 3D reconstruction tasks [63]. We leverage pretrained geometric priors from foundation models to achieve a coarse alignment of 3D point clouds $\mathcal{P}$ and camera parameters $\mathcal{C}$, promoting a stable and efficient learning process. For simplicity and stability in our pipeline, we adopt a simple backbone VGGT [72]. Specifically, given a pair of input frames $I^0, I^1$, the backbone outputs the corresponding image features $\mathcal{F}^0, \mathcal{F}^1$, 3D point clouds $\mathcal{P}^0, \mathcal{P}^1$, and camera parameters $\mathcal{C}^0, \mathcal{C}^1$.

**Quadratic Gaussian Head.** Real-world object surfaces are complex, making it difficult for point clouds to capture their surface structure accurately. QGS [23] defines Gaussian distributions on a quadratic surface, which can smoothly transition between convex and concave shapes. This flexibility allows for more accurate modeling of complex object surfaces. To leverage this capability, we propose the Quadratic Gaussian Head, a module inspired by QGS, that transforms point cloud features into QGS-based 3D primitives, enabling more effective surface representation. This surface representation enables estimating surface properties in subsequent stages, thereby preserving surface consistency across frames. Specifically, QGS contains the following parameters

$$\mathcal{G}^0 = \{\mu_i^0, o_i^0, r_i^0, s_i^0, c_i^0\}_{i=1,\dots,H \times W}, \quad (9)$$

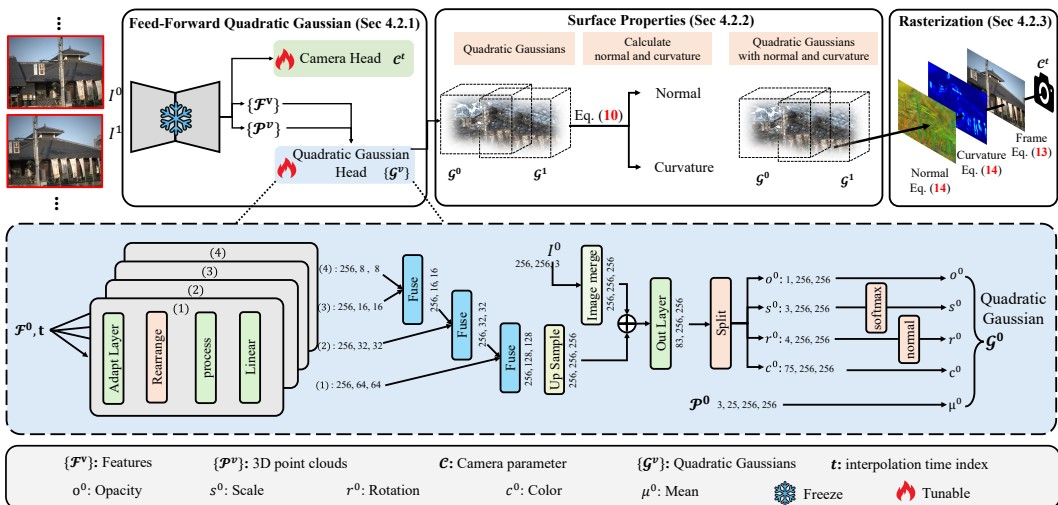

Figure 3: **Surface-Aware Feed-Forward Quadratic Gaussian** Framework. The Feed-Forward Quadratic Gaussian module transforms the input frames $I^0$ and $I^1$ into Quadratic Gaussian ($\mathcal{G}^0$ and $\mathcal{G}^1$) that represent surfaces. Then, the Surface Properties module computes the normal and curvature from the Quadratic Gaussian. Finally, the normal, curvature, and frame $I^t$ are rasterized onto the camera plane at the interpolation time.

where opacity $o$, rotation $r$, scale $s$ and color $c$ and $H \times W$ pixel numbers. Subsequently, the point cloud $\mathcal{P}^0$ is the QGS's mean $\mu$. Together with the predicted parameters, it forms the Quadratic Gaussian representation $\mathcal{G}_0$ of frame $I^0$.

### 4.2.2 Surface Properties

Normals usually play a crucial role in facilitating surface alignment and continuity across frames [73, 74]. Meanwhile, curvature characterizes the degree of surface bending [75, 76]. In regions with large motion, i.e., highly curved surface areas, relying solely on normals may lead to inaccurate alignment [77, 78]. This motivates employing higher-order geometric descriptors, such as curvature, to complement normal in fine-grained alignment surfaces. Therefore, we jointly utilize normals and curvature as surface constraints to ensure accurate and consistent alignment across frames.

Given a Quadratic Gaussian, we directly compute the normal and curvature at any point on the surface. Specifically, given any 3D point $\mathbf{p}$ on the surface [23], the corresponding normal and curvature are:

$$\mathbf{n}(\mathbf{p}) = \left(2\lambda_x x, 2\lambda_y y, -\frac{1}{s_3}\right), \quad \mathbf{k}(\mathbf{p}) = \frac{4\lambda_x \lambda_y}{\left(1 + 4\lambda_x^2 x^2 + 4\lambda_y^2 y^2\right)^2}, \tag{10}$$

where $\lambda_x = \frac{d_x}{s_1^2}$, and $d_x \in \{-1, 0, 1\}$ determines whether the paraboloid is elliptic, hyperbolic, or planar. More specifically, the $d_x$ idepends on both the positional variable $x$ and a temporal variable $t$, and is defined as $d_x = \tanh(t)\exp(x)$. For the detailed computation, please refer to Appendix A.1. Formally, the calculation process is written as:

$$\left\{\mathcal{G}^0, \mathcal{G}^1\right\} \mapsto \left\{N^0, K^0, N^1, K^1\right\}, \tag{11}$$

where $N$ and $K$ represent the normal map and the curvature map, respectively.

### 4.2.3 Rasterization

Finally, we rasterize the surface properties from the 3D space into the camera plane with the interpolation camera parameter. The total process can be written as:

$$\left\{\mathcal{G}^0, \mathcal{G}^1, N^0, K^0, N^1, K^1, \mathcal{C}^t\right\} \mapsto \{I^t, N^t, K^t\}, \tag{12}$$

where $\mathcal{C}^t = \mathcal{C}^0 \times t + \mathcal{C}^1 \times (1 - t)$ is the interpolation camera parameter.

Table 1: Quantitative comparison with SOTA methods on the standard benchmark, regarding PSNR/SSIM. The best and the second best results are denoted by pink and yellow.

| | Vimeo-90K [13] | UCF101 [79] | SNU-FILM [80] | | | | Average |
|---|---|---|---|---|---|---|---|
| | | | easy | medium | hard | extreme | |
| DAIN[25] | 34.71/0.9756 | 34.39/0.9683 | 39.73/0.9902 | 35.46/0.9780 | 30.17/0.9335 | 25.09/0.8584 | 33.36/0.9507 |
| AdaCoF[40] | 34.47/0.9730 | 34.90/0.9680 | 39.80/0.9900 | 35.05/0.9754 | 29.46/0.9244 | 24.31/0.8439 | 33.00/0.9458 |
| CAIN[80] | 34.65/0.9730 | 34.91/0.9690 | 39.89/0.9900 | 35.61/0.9776 | 29.90/0.9270 | 24.78/0.8507 | 33.29/0.9493 |
| Softsplat[27] | 36.13/0.9805 | 35.17/0.9690 | 40.26/0.9911 | 36.09/0.9798 | 30.93/0.9365 | 25.16/0.8608 | 33.92/0.9530 |
| XVFI[17] | 35.09/0.9759 | 35.17/0.9685 | 39.93/0.9907 | 35.37/0.9776 | 29.58/0.9276 | 24.17/0.8450 | 33.22/0.9477 |
| M2M-VFI[21] | 35.20/0.9768 | 35.28/0.9697 | 40.10/0.9906 | 36.12/0.9797 | 30.63/0.9368 | 25.27/0.8601 | 33.68/0.9519 |
| RIFE[15] | 35.61/0.9779 | 35.29/0.9697 | 40.10/0.9906 | 36.12/0.9797 | 30.63/0.9368 | 25.27/0.8601 | 33.68/0.9519 |
| IFRNet-L[81] | 36.20/0.9808 | 35.42/0.9698 | 40.36/0.9910 | 36.12/0.9797 | 30.63/0.9368 | 25.27/0.8609 | 33.96/0.9531 |
| AMT-L[82] | 36.35/0.9815 | 35.39/0.9696 | 39.95/0.9913 | 36.09/0.9805 | 30.75/0.9384 | 25.41/0.8638 | 33.99/0.9542 |
| EMA-VFI-S[20] | 36.64/0.9819 | 35.48/0.9701 | 39.98/0.9905 | 36.09/0.9801 | 30.94/0.9392 | 25.69/0.8661 | 34.14/0.9547 |
| SGM-VFI[12] | 35.81/0.9793 | 35.40/0.9693 | 40.14/0.9907 | 36.06/0.9795 | 30.81/0.9375 | 25.69/0.8661 | 33.96/0.9535 |
| VFIMamba-S[29] | 36.09/0.9800 | 35.36/0.9696 | 40.21/0.9909 | 36.17/0.9800 | 30.80/0.9381 | 25.59/0.8655 | 34.14/0.9540 |
| Ours | 36.06/0.9791 | 35.40/0.9692 | 39.98/0.9906 | 36.10/0.9798 | 30.90/0.9391 | 25.50/0.8651 | 33.35/0.9475 |

Table 2: Quantitative comparison with VFI methods on large motion benchmark. The best and the second best results are denoted by pink and yellow.

| | X-Test-L [12] | | SNU-FILM-L [12] | | Xiph-L [12] | | Runtime (s) | FLOPs (T) |
|---|---|---|---|---|---|---|---|---|
| | 2K | 4K | hard | extreme | 2K | 4K | | |
| XVFI[17] | 29.82/0.8951 | 29.02/0.8866 | 27.58/0.9095 | 22.99/0.8260 | 29.17/0.8449 | 28.09/0.7889 | 0.075 | 0.37 |
| FILM[18] | 30.08/0.8941 | 29.10/0.8886 | 28.35/0.9156 | 23.06/0.8247 | 29.89/0.8533 | 27.11/0.7699 | 1.29 | 1.36 |
| BiFormer[19] | 30.32/0.9067 | 30.15/0.9070 | 28.18/0.9154 | 23.85/0.8393 | 29.61/0.8541 | 28.98/0.8183 | 1.09 | 0.39 |
| RIFE[15] | 29.87/0.8805 | 28.98/0.8756 | 28.19/0.9172 | 22.84/0.8230 | 30.18/0.8633 | 28.07/0.7982 | 0.20 | 0.2 |
| AMT-L[82] | 29.39/0.8771 | 28.35/0.8731 | 28.33/0.9184 | 23.14/0.8288 | 30.32/0.8710 | 28.27/0.8095 | 0.58 | 0.58 |
| EMA-VFI-S[20] | 29.51/0.8775 | 28.60/0.8733 | 28.57/0.9189 | 23.18/0.8292 | 30.54/0.8718 | 28.40/0.8109 | 0.076 | 0.91 |
| SGM-VFI[12] | 30.39/0.8946 | 29.25/0.8861 | 28.90/0.9209 | 23.19/0.8301 | 30.89/0.8745 | 28.59/0.8115 | 0.93 | 1.79 |
| VFIMamba-S[29] | 31.58/0.9169 | 30.50/0.9077 | 28.80/0.9208 | 23.41/0.8300 | 30.72/0.8780 | 28.62/0.8111 | 0.128 | 0.24 |
| Ours | 31.33/0.9011 | 30.13/0.9066 | 29.05/0.9213 | 24.20/0.8400 | 31.20/0.8814 | 29.19/0.8197 | 0.340 | 1.28 |

Specifically, the rasterization of color is performed according to the following equation:

$$I^t(u,v) = \sum_{i=0}^{N-1} \mathbf{c}_i \alpha_i^{2D} \prod_{j=0}^{i-1}(1 - \alpha_i^{2D}). \tag{13}$$

Similarly, [23] renders the normal and curvature on the camera plane as follows:

$$N^t(u,v) = \sum_{i=0}^{N-1} \mathbf{n}_i \alpha_i^{2D} \prod_{j=0}^{i-1}(1 - \alpha_i^{2D}), \quad K^t(u,v) = \sum_{i=0}^{N-1} \mathbf{k}_i \alpha_i^{2D} \prod_{j=0}^{i-1}(1 - \alpha_i^{2D}), \tag{14}$$

where $N$ denotes the pixel number of the rendered image.

### 4.3 Loss Function

Finally, we minimize the following loss function:

$$\mathcal{L} = \mathcal{L}_c + \alpha\mathcal{L}_{kn}, \quad \mathcal{L}_{kn} = (1 - \text{sigmoid}(\ln(|K(u,v)|) + \varepsilon))\mathcal{L}_n \tag{15}$$

where $\mathcal{L}_c = \|I_{gt}^t - I^t\|_2$ is an RGB reconstruction loss function. $\mathcal{L}_{kn}(u,v)$ denotes the curvature-aware normal loss, which enforces surface alignment [23]. And $\mathcal{L}_n$ [71] is the normal consistency loss to ensure primitives are locally aligned with the surface.

## 5 Experiments

**Metrics.** We use common quantitative metrics: Peak Signal-To-Noise Ratio (PSNR) and Structural Similarity Image Metric (SSIM), where higher scores indicate better image quality. To assess temporal consistency between frames, we additionally employ the tOF metric [17].

To further illustrate the effectiveness of our algorithm in addressing large motion scenarios, we present a statistical analysis of the relationship between motion magnitudes [8, 18] and the corresponding

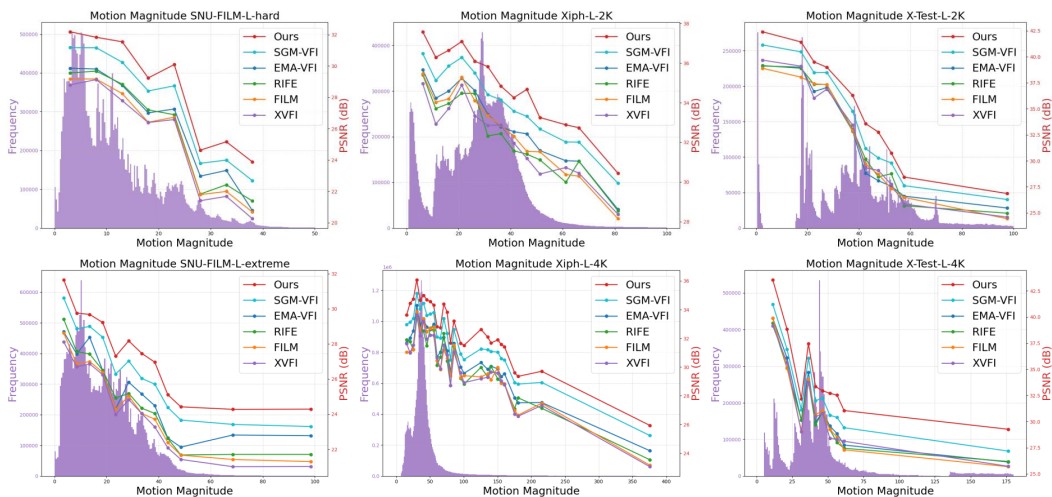

Figure 4: PSNR versus motion magnitude. Higher motion magnitudes correspond to larger inter-frame displacements, representing more challenging motion scenarios.

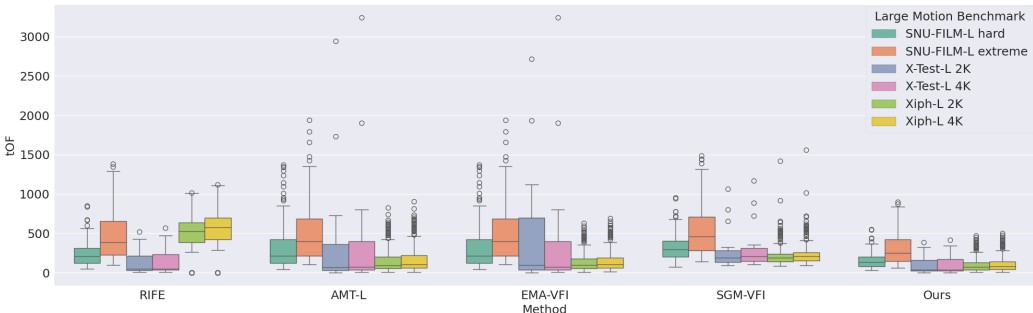

Figure 5: Comparison of temporal consistency under the tOF metric.

PSNR performance, as shown in Figure 4. Note that a higher motion magnitude corresponds to larger inter-frame displacement, indicating more challenging motion scenarios.

**Datasets.** For fair comparison, we follow the training and testing datasets established by the large motion benchmark [12]. For training, we follow the setting [12], utilizing both the Vimeo90K and X4K1000FPS (X-Train) datasets. Vimeo90K [13] consists of 51,312 triplets with a resolution of 448×256, featuring an average motion magnitude between 1 and 8 pixels. X4K1000FPS (X-Train) [17] contains 4,408 clips at a resolution of 768×768, with each clip comprising 65 consecutive frames.

We evaluate its performance following the large motion benchmark introduced by SGM-VFI [12]. X-Test-L [17, 12] with the largest temporal gap, as our primary benchmark for evaluating large motion scenarios. We also choose the 0th and 32nd frames as input and evaluate the quality of the synthesized 16th output frame. SNU-FILM-L [80, 12] is the most challenging half of the SNU-FILM hard and extreme, with 155 triplets each. Xiph-L [12] is constructed based on the original Xiph dataset [83] by doubling the input temporal intervals and retaining the most challenging half of the data to form this benchmark.

**Implementation Detail.** We optimize the loss using Adam in PyTorch framework. The cosine scheduler schedules the learning rate from $1e-4$ to $1e-6$. Standard data augmentation techniques, such as flipping, rotation, and cropping, are applied to the data with a size of $518 \times 280$. We train our model on the training datasets with a batch size 16 for 800 epochs.

## 5.1 Comparison with Previous Methods

As shown in the Table 1 and 2, methods are compared, which are tested on the standard and large motion benchmark. To comprehensively evaluate the capability of our model, we conduct experiments on benchmarks with varying motion magnitudes. We compare our method against recent video frame

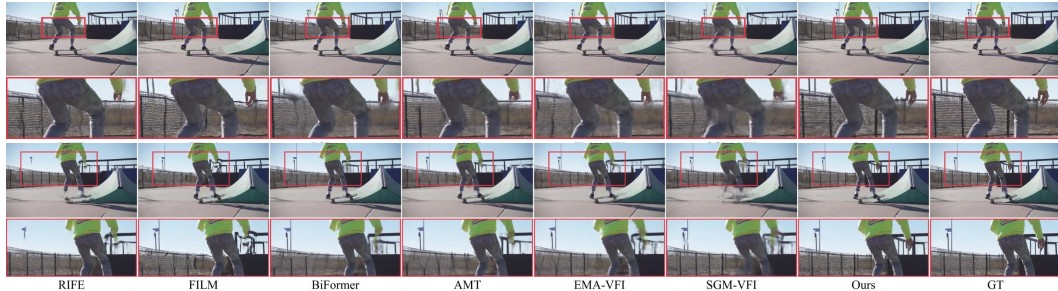

| RIFE | FILM | BiFormer | AMT | EMA-VFI | SGM-VFI | Ours | GT |

Figure 6: Visual comparison with state-of-the-art methods.

interpolation (VFI) approaches, including those specifically designed for large motion, as well as methods that perform well on standard benchmarks.

**Large Motion.** The large motion benchmark contains a significant number of scenes with large inter-frame displacements, resulting in motion magnitudes that are higher than those in standard benchmarks. As shown in Table 2, our method consistently outperforms state-of-the-art approaches on the large motion benchmark, demonstrating its effectiveness in handling complex motion scenarios. To further analyze performance under varying motion magnitude, we examine the PSNR across different motion magnitude intervals, as illustrated in Figure 4.

**Comparison of Temporal Consistency.** We employ the tOF metric [17] to evaluate the temporal consistency. As shown in Figure 5, our method consistently outperforms existing approaches on the large motion benchmark in terms of both the mean and variance of tOF, indicating more stability. This superior temporal consistency can be attributed to our method's accurate modeling of surface properties, which enables fine alignment of object correspondences across frames. We employ the tOF metric [17] to evaluate the temporal consistency. Figure 5 reports the quality of motion reconstruction across several existing models on the large motion benchmark. We can clearly observe that our method consistently outperforms existing methods in both the mean and variance of tOF. Outstanding temporal consistency is due to our method's fine alignment of object correspondence by modeling the surface properties, which improves motion temporal consistency.

**Comparison of Visual Results.** We further compare the visualization results in large motion. Figure 6 compares our method and several state-of-the-art approaches. CNN-based methods estimate pixel-level object correspondences, which often fall into local optima under large motion, leading to subpar interpolation results. Attention-based methods estimate patch-level object correspondences, which improves interpolation results under large motion.

## 5.2 Ablation Study

In this section, we present experimental insights to analyze and discuss the questions raised in the previous section: which 3D primitives are suitable for representing differential surfaces, and which surface properties facilitate surface alignment.

**Architecture.** To demonstrate the superiority of the QGS head in capturing complex geometric structures compared to the 3DGS head, we first compare their texture representations. Figure 7 provides qualitative evidence, while Table 3 offers quantitative validation of the QGS head's effectiveness.

Table 3: Ablation studies of Architecture on SNU-FILM-L extreme.

| Setting | Backbone | | | Gaussian Head | | Perfermence | |
|---------|----------|---------|------|----------|----------|------|--------|
| | CUT3R | MonST3R | VGGT | 3DGS head | QGS head | PSNR | tOF(↓) |
| (i) | ✓ | | | ✓ | | 23.27 | 308 |
| (ii) | ✓ | | | | ✓ | 24.11 | 240 |
| (i) | | ✓ | | ✓ | | 23.25 | 297 |
| (ii) | | ✓ | | | ✓ | 24.12 | 236 |
| (i) | | | ✓ | ✓ | | 23.30 | 286 |
| (ii) | | | ✓ | | ✓ | 24.20 | 226 |

Figure 7 presents a visual comparison between the two heads, highlighting the QGS head's ability to preserve fine-grained geometric details. As shown in the error maps of Figure 7, the surface-aware QGS head achieves significantly better reconstruction quality, particularly in regions with complex textures.

To further isolate the contribution of the QGS head from that of the backbone, we conduct an ablation study by combining different backbones, CUT3R [84], VGGT [72], and MonST3R [85], with both the 3DGS and QGS heads. Table 3 summarizes the performance of these combinations. The results show that the QGS head consistently outperforms all other configurations, especially under challenging

large motion scenarios, demonstrating its effectiveness in improving both reconstruction quality and temporal consistency.

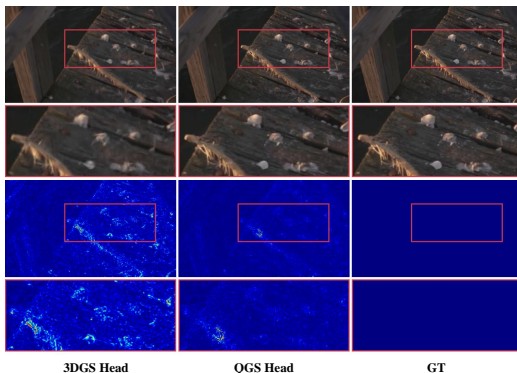

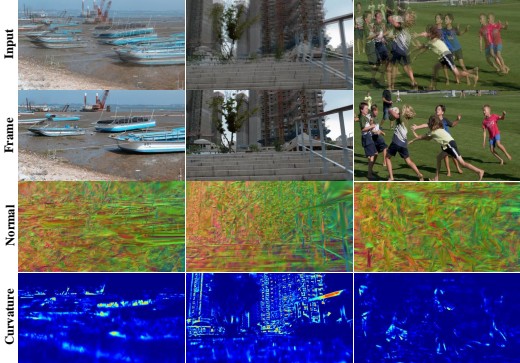

Figure 7: A visual comparison between the 3DGS head and the QGS head. The bottom part shows a heatmap of the interpolated frame error.

Figure 8: We visualize the surface properties, and notably, as discussed earlier, the curvature highlights motion regions, particularly those corresponding to rapidly bending surfaces.

**Surface properties.** We conduct an ablation study to further investigate the role of surface properties in enhancing surface alignment. Both qualitative and quantitative results are presented in Figure 8 and Table 4. Figure 8 visualizes the surface normals and curvature maps rendered by the QGS head. Notably, the curvature map highlights regions with high surface variation, such as bends and folds. This observation supports our earlier analysis that curvature serves as a higher-order geometric descriptor, complementing surface normals and facilitating more accurate surface alignment. These additional insights contribute to improved video frame interpolation performance in large motion.

Table 4: Ablation studies of Surface properties on X-Test-L 2K.

| | Surface Properties | | | Perfermence | |
|---------|-----|--------|-----------|------|--------------|
| Setting | RGB | Normal | Curvature | PSNR | tOF ($\downarrow$) |
| (i)     | ✓   |        |           | 29.07 | 183 |
| (ii)    | ✓   | ✓      |           | 30.47 | 117 |
| (iii)   | ✓   | ✓      | ✓         | 31.33 | 96 |

## 6 Conclusion

This work is the first to analyze frame-level object correspondence under large motion from the perspective of differential surface. Building on this insight, we propose an explicit Surface-Aware Feed-Forward Quadratic Gaussian pipeline to mitigate the challenge. Specifically, the proposed method transforms video frames into Quadratic Gaussians representing differential surfaces. Within this representation, we compute corresponding surface properties, such as normal and curvature. These properties are rendered onto the camera plane for explicit supervision and alignment. Extensive experiments demonstrate that our framework achieves state-of-the-art performance on the large motion benchmark, highlighting its effectiveness and robustness in handling complex motion scenarios. This framework opens new avenues for incorporating differential surface into the video frame interpolation task, particularly under large motion conditions.

**Limitation.** While our pipeline can cover most cases of large motion, there are many other cases beyond that coverage. The main reason for the limitations is that our definition of large motion and the proposed ideas are somewhat naive, which makes the solution sub-optimal for geometry. Our current definition focuses more on static correspondences in the background regions across different frames. For dynamic correspondences, due to the relatively short time interval between adjacent frames, we adopt a simplified assumption of linear motion in this work. At present, we employ a relatively basic differential surface theory to model the problem. We believe that, in the future, a more unified modeling of camera motion and object motion within a comprehensive differential geometry framework could lead to a more accurate characterization of complex dynamic scenes.

**Acknowledgment.** This work was supported by the National Key R&D Program of China (2022ZD0161800), the National Natural Science Foundation of China under Grant 62271203, AI-Empowered Research Paradigm Reform and Discipline Leap Plan under Grant 2024AI01012 and the Open Research Fund of KLATASDS-MOE, ECNU.

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

# A  Theoretical Supplement

## A.1  Surface Properties of Quadratic Gaussian

A paraboloid is defined as:

$$f(x, y, z) = \begin{bmatrix} x & y & z & 1 \end{bmatrix} \begin{bmatrix} \frac{d_x}{s_1^2} & 0 & 0 & 0 \\ 0 & \frac{d_y}{s_2^2} & 0 & 0 \\ 0 & 0 & 0 & -\frac{d_z}{2s_3} \\ 0 & 0 & -\frac{d_z}{2s_3} & 0 \end{bmatrix} \begin{bmatrix} x \\ y \\ z \\ 1 \end{bmatrix}$$

$$= \frac{d_x}{s_1^2}x^2 + \frac{d_y}{s_2^2}y^2 - \frac{d_z}{s_3}z = 0. \tag{16}$$

$\mathbf{S} = diag(s_1, s_2, s_3)$, which denotes the orientation and scale of the quadric in the object space. The matrix $D$ defines the surface shape: $\mathbf{D} = diag(1, 1, 1, 1)$ yields an ellipsoid, while $\mathbf{D} = diag(1, 0, 0, 0)$ produces a plane. $d_{ii} \in \{0, \pm 1\}$ determines whether the paraboloid is elliptic, hyperbolic, or planar. For convenience in writing and subsequent derivations, we simplify Equation 16 as follows:

$$f(x, y, z) = \lambda_x x^2 + \lambda_y y^2 - \frac{1}{s_z}z = 0 \tag{17}$$

### A.1.1  Normal

QGS is a surface-based representation that naturally possesses multiview consistent geometric properties, making it straightforward to compute surface normals. Given any point $\mathbf{p} = (x, y, z)$ on the surface, we can take the partial derivatives of Eq. 17, yielding:

$$\mathbf{n(p)} = \left( 2\lambda_x x, 2\lambda_y y, -\frac{1}{s_z} \right), \tag{18}$$

### A.1.2  Curvature

Here, we compute the Gaussian curvature analytically using a standard differential geometry approach [22]. By the way, throughout the entire paper, the parameter domain is expressed using $(u, v)$ coordinates, while the surface is represented using $(x, y, z)$ coordinates. We simplify Eq. 17 as $z = \lambda_x x^2 + \lambda_y y^2$. Given the point $\mathbf{p} = (x, y, z)$, the partial derivatives are:

$$x_u = (1, 0, 2\lambda_x x) \tag{19}$$
$$x_v = (0, 1, 2\lambda_y y) \tag{20}$$

The first fundamental form is:

$$E = \langle x_u, x_u \rangle = 1 + 4\lambda_x^2 x^2 \tag{21}$$
$$F = \langle x_u, x_v \rangle = 4\lambda_x \lambda_y xy \tag{22}$$
$$G = \langle x_v, x_v \rangle = 1 + 4\lambda_y^2 y^2 \tag{23}$$

The second fundamental form is:

$$n = \frac{x_u \times x_v}{\|x_u \times x_v\|} = \frac{(-2\lambda_x x, -2\lambda_y y, 1)}{\sqrt{1 + 4\lambda_x^2 x^2 + 4\lambda_y^2 y^2}} \tag{24}$$

$$x_{uu} = (0, 0, 2\lambda_x) \tag{25}$$
$$x_{uv} = (0, 0, 0) \tag{26}$$
$$x_{vv} = (0, 0, 2\lambda_y) \tag{27}$$

$$L = \langle n, x_{uu} \rangle = \frac{2\lambda_x}{\sqrt{1 + 4\lambda_x^2 x^2 + 4\lambda_y^2 y^2}} \tag{28}$$

$$M = \langle n, x_{uv} \rangle = 0 \tag{29}$$

$$N = \langle n, x_{vv} \rangle = \frac{2\lambda_y}{\sqrt{1 + 4\lambda_x^2 x^2 + 4\lambda_y^2 y^2}} \tag{30}$$

Finally, the Gaussian curvature can be computed as:

$$\mathbf{k}(\mathbf{p}) = \frac{LN - M^2}{EG - F^2} = \frac{\dfrac{4\lambda_x\lambda_y}{1 + 4\lambda_x^2 x^2 + 4\lambda_y^2 y^2}}{1 + 4\lambda_x^2 x^2 + 4\lambda_y^2 y^2} = \frac{4\lambda_x\lambda_y}{\left(1 + 4\lambda_x^2 x^2 + 4\lambda_y^2 y^2\right)^2} \tag{31}$$

## B  More Visual Results

The anonymous GitHub repository provides visualization results in both video and 3D formats.

