# OpenReview forum: "Surface-Aware Feed-Forward Quadratic Gaussian for Frame Interpolation with Large Motion"
_NeurIPS.cc/2025/Conference — NeurIPS 2025 poster_

### Official Review · Reviewer_aERN · 2025-06-26

**Clarity:** 2
**Significance:** 3
**Originality:** 3
**Rating:** 5
**Confidence:** 3

**Summary:**

The authors tackle the task of video frame interpolation by predicting Quadratic Gaussians from the two input frames (along with relative pose). They then project the estimated Gaussians on the target view (with pose given by interpolation under a constant velocity assumption). Performance is measured across a wide variety of datasets, showing a consistent improvement over existing methods.

**Questions:**

- How is the method able to handle dynamic motions, such as the skateboarding person in Figure 6? A discussion around this in the paper would be informative.
- How does the proposed method relate to [1, 2] (see s&w section)?

**Ethical Concerns:**

["NO or VERY MINOR ethics concerns only"]

**Final Justification:**

The authors have addressed my comments and provided additional baselines. I will raise my score to accept

**Limitations:**

I wonder what is meant by “the limitations is that our definition of large motion and the proposed ideas are somewhat naive”. Could you explain further?

**Paper Formatting Concerns:**

- Should be plural “Quadratic Gaussians”, not “Quadratic Gaussian” for most places in the text and title
- L234: missing reference link to Table 2

**Quality:**

3

**Strengths And Weaknesses:**

Strengths:
- The proposed approach is elegant and well motivated
- Evaluation is extensive and covers a wide range of datasets, with varying settings and difficulty levels.
Ablation experiments are well structured and well motivates the pipeline architecture.

Weaknesses:
- Lacking comparison to prior works [1,2], which both predict regular Gaussians and poses
- The connection between Thm 1 and the method is not very clear. For example, it is not explained what “fundamental forms” are. Please add some sentences to draw the connection to surface curvature and normals.
- Including eq.7 (or similar) from the QGS paper would be helpful to make the background more self contained and make it easier to keep track of notations.


[1] Smart, Brandon, et al. "Splatt3r: Zero-shot gaussian splatting from uncalibrated image pairs." arXiv preprint arXiv:2408.13912 (2024).
[2] Ye, Botao, et al. "No pose, no problem: Surprisingly simple 3d gaussian splats from sparse unposed images." arXiv preprint arXiv:2410.24207 (2024).

---

> ### Author Rebuttal · Authors · 2025-07-30
>
> Thank you for your insightful inquiries, and we will provide detailed responses to each of them below:
>
> ---
>
> ## [W1] comparison to Splatt3r and No pose
>
> We thank the reviewer for the insightful question. We have conducted a comprehensive quantitative evaluation and visual comparison of the two methods. Specifically, we performed quantitative experiments on both the standard benchmark and the large motion benchmark, and the corresponding visual results are provided on the anonymous website linked in the appendix.
>
> **Splatt3r** is the first to propose a feed-forward Gaussian rendering pipeline. However, it suffers from significant computational overhead and resource consumption, which limits its scalability in practical applications.
>
> **Nopose** builds upon this by introducing a camera-space-based feed-forward Gaussian pipeline, achieving notable improvements in efficiency. However, its use of a 3D Gaussian head introduces limitations in handling fine geometric alignment.
>
>
> **Stand Benchmark**
> | Method    | SNU-FILM easy   | SNU-FILM medium | SNU-FILM hard  | SNU-FILM extreme | Runtimes (s) | FLOPs (T) |
> |-----------|------------------|------------------|-----------------|--------------------|----------------|------------|
> | Splatt3r  | 39.23 / 0.986   | 35.20 / 0.971   | 29.87 / 0.930  | 25.00 / 0.860     | 6.17           | 5.8        |
> | Nopose    | 39.97 / 0.990   | 36.00 / 0.979   | 30.47 / 0.937  | 25.43 / 0.865     | 0.39           | 1.9        |
>
>
> **Large Motion Benchmark**
> | Method    | X-text-L 2K     | X-text-L 4K     | SNU-FILM-L hard | SNU-FILM-L extreme | Xiph-L 2K     | Xiph-L 4K     |
> |-----------|------------------|------------------|------------------|---------------------|----------------|----------------|
> | Splatt3r  | 29.57 / 0.880   | 28.95 / 0.876   | 28.43 / 0.897   | 23.35 / 0.825       | 29.88 / 0.861 | 28.10 / 0.807 |
> | Nopose    | 30.17 / 0.895   | 29.27 / 0.886   | 28.71 / 0.920   | 23.51 / 0.832       | 30.57 / 0.873 | 28.80 / 0.811 |
>
> ---
>
> ## [W2] The connection between Thm 1 and the method is not very clear.
>
> We thank the reviewer for the valuable suggestion, which has helped improve the readability of our paper. In response, we provide the following clarification regarding Theorem 1:
>
> The **First Fundamental Form** is defined on the tangent space of a surface and is used to describe intrinsic geometric properties such as length, angle, and area. In contrast, the **Second Fundamental Form** characterizes the extrinsic curvature of the surface in 3D space， i.e., how the surface bends relative to its tangent plane， and is typically associated with geometric quantities such as **Gaussian curvature**.
>
> When establishing correspondences of objects across frames, **normal** is commonly used as a means of geometric alignment. However, in regions with significant geometric variation (e.g., object boundaries or areas with large motion), relying on normal constraints can introduce errors, often resulting in over-smoothing and degraded reconstruction quality in these areas.
>
> To address this issue, we introduce **curvature** as a descriptor of local surface bending. Curvature provides a more effective way to focus on regions with significant geometric changes and offers stronger supervision signals in these regions.
>
> ---
>
> ## [W3] QGS background.
>
> We thank the reviewer for the valuable feedback.
>
> Due to space limitations, the detailed computation of QGS was originally placed in Appendix A of the main text. In response to the reviewer’s suggestion, we will revise the structure of the paper and move the core content of QGS to the Preliminary section in the next version. This adjustment will help readers better understand the background and motivation of our proposed method.
>
> Here, we briefly introduce the definition of **QGS** as well as the computation of **normal** and **curvature**.
>
> **Definition**  A paraboloid (Quadratic Gaussian) is defined as
>
> $f(x,y,z) = \frac{d_{x}}{s_x^2} x^2 + \frac{d_{y}}{s_y^2} y^2 - \frac{d_{z}}{s_z} z = 0$.
>
> $\mathbf{S} = diag(s_x, s_y, s_z)$, which denotes the orientation and scale of the quadric in the object space. The matrix $\mathbf{D}$ defines the surface shape: $\mathbf{D} = diag(1, 1, 1, 1)$ yields an ellipsoid, while $\mathbf{D} = diag(1, 0, 0, 0)$ produces a plane.
> $d_{ii} \in \{0, \pm 1\}$ determines whether the paraboloid is elliptic, hyperbolic, or planar.
> For convenience in writing and subsequent derivations, we simplify the paraboloid as follows:
>
> $f(x, y, z) = \lambda_x x^2 + \lambda_y y^2 - \frac{1}{s_z} z = 0$.
>
> **Normal** Given any point $\mathbf{p} = (x, y, z)$ on the paraboloid, we can take the partial derivatives yielding: $\mathbf{n} ( \mathbf{p} ) = \left( 2 \lambda_x x,
> 2 \lambda_y y, -\frac{1}{s_z}\right).$
>
> **Curvature** We compute the Gaussian curvature analytically using a standard differential geometry approach
> $\mathbf{k}(\mathbf{p}) = \frac{LN - M^2}{EG - F^2}  =  \frac{4 \lambda_x \lambda_y}{\left(1 + 4 \lambda_x^2 x^2 + 4 \lambda_y^2 y^2\right)^2}$, where E, F, and  G are the first fundamental form, and L, M, and  N are the second fundamental form.
>
> ---
>
> ## [Q1] How is the method able to handle dynamic motions?
>
> This is a good question. We thank the reviewer for raising it. The dynamic motions follow the pipeline outlined:
>
> **(I)** Input Frames: We take two input video frames, $I^0 \in \mathbb{R}^{H \times W}$ and $I^1 \in \mathbb{R}^{H \times W \times 3}$, where  H and W denote the height and width of the image, respectively.
>
> **(II)**  Point Cloud: Frame correspondences can be categorized into two types: **static correspondences**, which are primarily caused by background changes due to camera motion, and **dynamic correspondences**, which result from object motion.
> VGGT demonstrates strong performance in modeling **static correspondences**.
> We leverage this capability to obtain the point clouds $\mathcal{P}^0 \in \mathbb{R}^{HW \times 3}$  and $\mathcal{P}^1 \in \mathbb{R}^{HW \times 3}$  aligned based on static correspondences, where $HW$ is the number of points (pixels), $3$ represents the 3D spatial coordinates.
> In addition, we visualize the scene by constructing $[ \mathcal{P}^0 || \mathcal{P}^1 ]$ to verify the quality of the static correspondence alignment in the Appendix( https://anonymous.4open.science/w/1018-B854/). Here, the symbol $||$  denotes the concatenation operation, i.e., torch.cat() in PyTorch.
>
> **(III)** Linear Motion:
> Due to the alignment of static correspondences and the short temporal interval between adjacent frames in video interpolation tasks,  we follow the linear motion assumption of previous works.
> Based on this, we compute the interpolated point cloud at time  $t \in [0, 1]$ as: $\mathcal{P}^t = (1-t) \mathcal{P}^0 + t \mathcal{P}^1$. By continuously varying $t$, we can generate a sequence of intermediate point clouds
> $\mathcal{P}^t \in \mathbb{R}^{HW\times3}$.
>
> **(IV)** Rendering Intermediate Frame: The interpolated point cloud $\mathcal{P}^t$  is then fed into the QGS head, which renders the intermediate frame $I^t$. We compute the loss between the rendered frame $I^t$  and the ground truth intermediate frame $I^t_{gt}$.
>
> To intuitively understand the motions, we have visualized the entire pipeline. The visualization results are provided in the Appendix (https://anonymous.4open.science/w/1018-B854/).
>
> ---
>
> ## [Limitation] I wonder what is meant by “the limitations are that our definition of large motion and the proposed ideas are somewhat naive”. Could you explain further?
>
> Frame correspondences can be categorized into two types: **static correspondences**, which are primarily caused by background changes due to camera motion, and **dynamic correspondences**, which result from object motion.
>
> Our current definition focuses more on **static correspondences** in the background regions across different frames. For **dynamic correspondences**, due to the relatively short time interval between adjacent frames, we adopt a simplified assumption of **linear motion** in this work.
>
> At present, we employ a relatively basic differential surface theory to model the problem. We believe that, in the future, a more unified modeling of camera motion and object motion within a comprehensive differential geometry framework could lead to a more accurate characterization of complex dynamic scenes.
>
> We sincerely hope that future research will further explore the integration of differential surface theory with feed-forward Gaussian rendering pipelines, which we believe holds great potential for advancing this field.
>
> *We hope our explanation and experiments address your concerns and demonstrate the improved quality of our paper! Thank you again for your valuable time!*

---

> > ### Comment · Reviewer_aERN · 2025-08-04
> >
> > Thank you for the detailed response and additional experiments.
> >
> > Regarding W2: I still don't see the connection between Thm 1 and the proposed loss formulation. Thm 1 suggests that we should learn to match the first and second fundamental forms with the ground truth surface. In contrast, the proposed loss minimizes a curvature-aware normal loss, which at first glance does not guarantee that the fundamental forms are matched. How are the two connected?

---

> ### Author Response · Authors · 2025-08-05
>
> ## [w2] The connection between the curvature-aware normal loss function and the first and second fundamental forms.
>
>
>
>
> We thank the reviewer for kind feedback and questions, which help us improve the quality of our manuscript.
>
> Ideally, if the normal vectors and curvatures between two differentiable surfaces are consistent, then their first and second fundamental forms must be identical[1].
>
> [1] Lipman, Y., & Daubechies, I. (2011). Conformal Wasserstein distances: Comparing surfaces in polynomial time. Advances in Mathematics, 227(3), 1047-1077.
>
> According to the fundamental theorem of surface theory, if two differentiable surfaces have the same first and second fundamental forms, they are congruent up to a rigid transformation.
> Our loss function aims to minimize the discrepancy of normal vectors and curvatures, which in turn encourages the alignment of these fundamental forms.
>
>
>
>
>
> A detailed explanation and proof are provided below. This proof will also be added to the appendix of our paper.
>
>
> **Definition**
>
> Let there be a rigid transformation between two differentiable surfaces given by
> $\tilde{X}(u, v) = R X(u, v) + T$, where $R$ is a rotation matrix and $T$ is a translation vector.
>
> For a surface  $X(u,v)$, its first fundamental form is:
>
> $I = E  du^2 + 2F  du  dv + G dv^2$
>
> The coefficients are defined as:
>
> $E = \langle X_u, X_u \rangle, \quad
> F = \langle X_u, X_v \rangle, \quad
> G = \langle X_v, X_v \rangle $
>
>
> The second fundamental form is:
>
> $II = Ldu^2 + 2Mdudv + Ndv^2$
>
>
> The coefficients are defined as:
>
> $L = \langle X_{uu}, N \rangle, \quad M = \langle X_{uv}, N \rangle$, $N = \langle X_{vv}, N \rangle$
>
> $N$ is the unit normal vector, defined as:
>
> $ N = \frac{X_u \times X_v}{\| X_u \times X_v \|} $
>
> **Proof I**
>
> Given the rigid transformation between the surfaces,
>
> $\tilde{X}(u,v) = RX(u,v) + T$.
>
> The first-order partial derivatives of the transformed surface $\tilde{X}(u,v)$ are:
>
> $\tilde{X}_u = R X_u, \quad \tilde{X}_v = R X_v$.
>
> Now, we compute $\tilde{E}, \tilde{F}, \tilde{G}$ of the second fundamental form for the transformed surface $\tilde{X}$.
>
> This holds because a rotation matrix R is an orthogonal transformation and thus preserves the inner product:
>
> $
> \langle Ra, Rb \rangle = \langle a, b \rangle \quad \forall a, b \in \mathbb{R}^3
> $
>
> Consequently, the first fundamental form of the transformed surface is identical to the original:
>
> $
> \tilde{I} = \tilde{E}  du^2 + 2\tilde{F} du dv + \tilde{G} dv^2
> = E du^2 + 2F dudv + Gdv^2 = I
> $
>
> This proves that the first fundamental form is invariant under rigid transformations.
> A similar proof holds for the invariance of the second fundamental form.
>
> **Proof II**
>
> Given the rigid transformation between the surfaces,
>
> $\tilde{X}(u,v) = RX(u,v) + T$.
>
> The first-order partial derivatives of the transformed surface $\tilde{X}(u,v)$ are:
>
> $\tilde{X}_u = R X_u, \quad \tilde{X}_v = R X_v$.
>
> The second-order partial derivatives are:
>
> $\tilde{X_{uu}} = R X_{uu},  \tilde{X_{uv}} = R X_{uv},  \tilde{X_{vv}} = R X_{vv}$
>
> The normal is:
>
> $
> \tilde{N} = \frac{\tilde{X}_u \times \tilde{X}_v}{\|\tilde{X}_u \times \tilde{X}_v\|}
> = \frac{RX_u \times RX_v}{\|RX_u \times RX_v\|}
> $
>
> Using the property of the vector cross product for a rotation matrix, we have:
>
> $
> RX_u \times RX_v = R(X_u \times X_v)
> $
>
> A rotation matrix preserves the norm of a vector:
>
> $
> \|R(X_u \times X_v)\| = \|X_u \times X_v\|
> $
>
> Therefore, the transformed normal vector is:
>
> $
> \tilde{N} = R N
> $
>
> Now, we compute  $\tilde{L}, \tilde{M}, \tilde{N}$ of the second fundamental form for the transformed surface $\tilde{X}$.
>
> $
> \tilde{L} = \langle \tilde{X_{uu}}, \tilde{N} \rangle
> = \langle R X_{uu}, R N \rangle
> = \langle X_{uu}, N \rangle = L,
> $
>
> $
> \tilde{M} = \langle \tilde{X_{uv}}, \tilde{N} \rangle
> = \langle R X_{uv}, R N \rangle
> = \langle X_{uv}, N \rangle = M,
> $
>
> $
> \tilde{N} = \langle \tilde{X_{vv}}, \tilde{N} \rangle
> = \langle R X_{vv}, R N \rangle
> = \langle X_{vv}, N \rangle = N.
> $
>
> Consequently, the second fundamental form of the transformed surface is identical to the original:
> $
> \tilde{II} = \tilde{L}  du^2 + 2\tilde{M}  du dv + \tilde{N} dv^2
> = L du^2 + 2M du dv + N dv^2 = II
> $
>
> This proves that the second fundamental form is invariant under rigid transformations.
>
>
> **Conclusion**
>
> Normal is:
> $
> N = \frac{X_u \times X_v}{\|X_u \times X_v\|}.
> $
> Curvature is:
> $
> K = \frac{LN - M^2}{EG - F^2},
> $
> where $E, F, G$ are the coefficients of the first fundamental form, and $L, M, N$ are the coefficients of the second fundamental form.
>
> If two differentiable surfaces have the same first and second fundamental forms, they are congruent up to a rigid transformation.
>
> Therefore, the loss function aims to minimize the discrepancy between normals and curvatures, thereby indirectly promoting the consistency of the two fundamental forms and ultimately achieving accurate surface alignment.
>
> *Once again, we thank the reviewer for the kind feedback.*

---

> > ### Comment · Reviewer_aERN · 2025-08-05
> >
> > I thank the authors for another detailed response. As you have shown, the normals and $K(u,v)$ can be written in terms of the fundamental forms. What I am missing is:  in what sense does minimizing $\mathcal{L}_{kn}$ ensure consistency between first and second fundamental forms?
> >
> > More concretely: if aligning fundamental forms is the goal, why not have a loss term directly comparing the rendered $E,F,G,L,M,N$ with their respective ground truth values? In contrast, $\mathcal{L}_{kn}$ in Eq. 9 is minimized by maximizing $K(u, v)$ and/or aligning the normals with the ground truth.
> >
> > Unless I am missing something, I would suggest replacing Thm 1 with a more intuitive geometric motivation along the lines of the first paragraph of Section 3.2.2.

---

> ### Author Response · Authors · 2025-08-06
>
> We sincerely thank the reviewer for the constructive suggestion.
> We are pleased to accept the reviewer's advice and improve the readability of the paper.
>
> **To enhance the readability and intuitive understanding of the paper, we have replaced Theorem 1 with an intuitive description as follows:**
>
> *In large motion scenes, video frame interpolation methods often fall into suboptimal object correspondences, as they operate at the pixel level or patch level and thus struggle to model frame-level object correspondences.
> To address the limitations of local object correspondences, we redefine the large motion problem as a global frame-level alignment task by aligning the surfaces that represent each frame.
> The core idea of frame-level alignment is that aligning the geometric properties between surfaces (such as normals and curvatures) can facilitate the alignment between surfaces, as illustrated in Figure 2.*
>
>
> *In the following sections, we describe how to construct surface representations from
> video frames and how to select surface properties to facilitate accurate correspondences across surfaces.*
>
>
> Please don't hesitate to reply if you have any further suggestions.
> We greatly appreciate your valuable suggestions, which we will incorporate into our revision to enhance the quality of the paper.
> Thank you again for your valuable time and constructive suggestions!

---

### Official Review · Reviewer_HLES · 2025-07-01

**Clarity:** 2
**Significance:** 3
**Originality:** 3
**Rating:** 4
**Confidence:** 3

**Summary:**

To address the challenges of current video frame interpolation (VFI) methods in handling large motion scenarios, this paper introduces a differential surface representation to model videos, aiming to capture frame-level object correspondences and geometric constraints. Specifically, the authors propose a Surface-Aware Feed-Forward Quadratic Gaussian (SA-FFQG) framework that maps video frames to 3D surfaces, thereby mitigating the limitations of local correspondences. Experimental results demonstrate strong performance on benchmarks with large motion magnitudes.

**Questions:**

Questions

1. Intuitive Explanation for Global Matching: Could the authors provide more intuitive evidence or visualizations to illustrate why Quadratic Gaussian Splatting (QGS) enables better global correspondences compared to traditional methods?

2. Comparison with VFIMamba: Please include a comparison with VFIMamba to validate the method's performance against the current state-of-the-art.

3. Visual Validation: Additional video visualizations, especially for challenging scenarios with large motions and non-rigid objects, are essential to demonstrate the method's generalization ability.

**Ethical Concerns:**

["NO or VERY MINOR ethics concerns only"]

**Final Justification:**

I thank the author for patiently addressing my questions, and I have decided to improve my score.

**Limitations:**

yes

**Paper Formatting Concerns:**

Not found.

**Quality:**

2

**Strengths And Weaknesses:**

Strengths

1. Novelty and Potential Impact: The proposed Feed-Forward Quadratic Gaussian framework exhibits significant novelty in the VFI task and has the potential to inspire future research directions.

2. Pioneering Use of Gaussian Splatting: As the first work to apply Gaussian Splatting in VFI, the method achieves competitive performance compared to prior state-of-the-art approaches.

Weaknesses

1. Clarity of Theoretical Contributions: As a non-expert in Gaussian Splatting, I found the mathematical details in Equations 4-8 challenging to follow. Specifically, I am unsure whether the superscript t in Equations 7 and 8 is necessary, as both equations appear to involve interpolations between two frames. Further clarification would enhance the accessibility of the work.

2. Computational Efficiency: Given the use of a pre-trained VGGT model, the paper does not explicitly report the total parameter count. Table 2 indicates that the proposed method is not outperformed in terms of speed, which may limit its practical applicability in real-time scenarios such as gaming.

3. Generalization to Non-Rigid Motion: Theorem 1 is derived under the assumption of rigid motion, whereas real-world scenarios often involve non-rigid deformations. It remains unclear how this assumption affects the method's generalization capability and practical performance.

4. Incomplete Baseline Comparison: The experiments lack a comparison with the current SOTA method VFIMamba on large motion benchmarks, which weakens the claim of superiority.

5. Limited Visualization: The paper provides few video examples and lacks qualitative visualizations, which undermines the credibility of the reported results.

---

> ### Author Rebuttal · Authors · 2025-07-30
>
> We appreciate your invaluable insights and thoughtful comments. In the following sections, we address the questions you have raised:
>
>  ---
>
> ## [W1] Explanation of the Gaussian Splatting formula
>
> We apologize for the confusion caused by the insufficient explanation. In the revised version, we will provide a more detailed description of **3D Gaussian Splatting**, along with clearer explanations of the meanings and roles of **Equations ($\textcolor{red}{4}$), ($\textcolor{red}{7}$), and ($\textcolor{red}{8}$)**.
>
> ### **Preliminary: 3D Gaussian Splatting**
>
> Specifically, 3DGS explicitly parameterizes Gaussian primitives via 3D covariance matrix  $\Sigma$ and their location point $\mathbf{p} _i(x,y,z)$, $i \in H\times W$, and $H\times W$ denotes the point number.
>
> $\mathcal{G}(\mathbf{p}) = \exp\left( -\frac{1}{2} ( \mathbf{p} -  \mathbf{p}_i)^\top \Sigma^{-1} ( \mathbf{p}  -  \mathbf{p}_i) \right)$,
>
>  where the covariance matrix $\Sigma = RSS^{\top}R^{\top}$ is factorized into a scaling matrix $S$ and a rotation matrix $R$.
>
> To render an image, the 3D Gaussian is transformed into the camera coordinates with world-to-camera transform matrix $W$ and projected to the image plane via a local affine transformation $J$:
>
> $\Sigma' = J W \Sigma W^\top J^\top$.
>
> By skipping the third row and column of $\Sigma'$, we obtain a 2D Gaussian $\mathcal{G}^{2D}$ with covariance matrix $\Sigma^{2D}$.
>
> Next, 3DGS employs volumetric alpha blending to integrate alpha-weighted appearance from front to back:
>
> $I^t(u, v) = \sum_{i}^{H\times W} \mathcal{G}_i^{2D}(u,v) o_i \mathbf{c}_i  \prod_j^{i-1} (1 - \mathcal{G}_j^{2D}(u,v) o_j)$
>
> where i is the index of the Gaussian primitives, $o_i$ denotes the alpha values, and $\mathbf{c}_i$ is the view-dependent appearance (color).
>
> ### **Explanation Equations ($\textcolor{red}{4}$), ($\textcolor{red}{7}$), and ($\textcolor{red}{8}$)**
>
> **(I)** The background-aligned point clouds $\mathcal{P}^0$ and $\mathcal{P}^1$ are linearly approximated to obtain the intermediate point cloud $\mathcal{P}^t$ using the formula $\mathcal{P}^t = (1-t)\mathcal{P}^0 + t\mathcal{P}^1$.
> The resulting point cloud $\mathcal{P}^t$  is then transformed into a Quadratic Gaussian $\mathcal{G}^t$  through the QGS head.
>
>  **(II)** Subsequently, the normal and curvature attributes are computed from $\mathcal{G}^t$ using Equation ($\textcolor{red}{4}$).  The detailed derivation of Equation ($\textcolor{red}{4}$) is provided in Appendix A of the main paper. Here, we briefly introduce the definition of QGS as well as the computation of normal and curvature about Equation ($\textcolor{red}{4}$).
>
>  **(II.i) Definition** A Quadratic Gaussian (paraboloid)  is defined as
>
> $f(x,y,z) = \frac{d_{x}}{s_x^2} x^2 + \frac{d_{y}}{s_y^2} y^2 - \frac{d_{z}}{s_z} z = 0$.
>
> $\mathbf{S} = diag(s_x, s_y, s_z)$, which denotes the orientation and scale of the quadric in the object space. The matrix $\mathbf{D}$ defines the surface shape: $\mathbf{D} = diag(1, 1, 1, 1)$ yields an ellipsoid, while $\mathbf{D} = diag(1, 0, 0, 0)$ produces a plane.
> $d_{ii} \in \{0, \pm 1\}$ determines whether the paraboloid is elliptic, hyperbolic, or planar.
> For convenience in writing and subsequent derivations, we simplify the paraboloid as follows:
>
> $f(x, y, z) = \lambda_x x^2 + \lambda_y y^2 - \frac{1}{s_z} z = 0$.
>
>  **(II.ii) Normal:** Given any point $\mathbf{p} = (x, y, z)$ on the paraboloid, we can take the partial derivatives yielding: $\mathbf{n} ( \mathbf{p} ) = \left( 2 \lambda_x x,
> 2 \lambda_y y, -\frac{1}{s_z}\right).$
>
>  **(II.iii) Curvature** We compute the Gaussian curvature analytically using a standard differential geometry approach
> $\mathbf{k}(\mathbf{p}) = \frac{LN - M^2}{EG - F^2}  =  \frac{4 \lambda_x \lambda_y}{\left(1 + 4 \lambda_x^2 x^2 + 4 \lambda_y^2 y^2\right)^2}$, where E, F, and  G are the first fundamental form, and L, M, and  N are the second fundamental form.
>
> **(III)** Based on Equation  ($\textcolor{red}{7}$) in the main paper, the color attribute $\mathbf{c}$ of the QGS $\mathcal{G}^t$ is projected onto the camera plane, resulting in the intermediate frame $I^t(u,v)$.
>
> **(IV)** Based on Equation  ($\textcolor{red}{8}$) in the main paper, the normal  $\mathbf{n}$ and curvature  $\mathbf{k}$ attributes of the QGS $\mathcal{G}^t$ is projected onto the camera plane, resulting the normal map $N^t(u,v)$ and curvature map $K^t(u,v)$.
>
> ---
>
> ## [W2]  Total parameter
>
> Thank you for pointing this out. We apologize for the omission. The total number of parameters in our model is 119.059M, and we will include this information in the revised manuscript.
>
> ---
>
> ## [W3 & Q3]  Generalization
>
> Frame correspondences can be categorized into two types: **static correspondences**, which are primarily caused by background changes due to camera motion, and **dynamic correspondences**, which result from object motion.
> Our method mainly focuses on handling large background displacements caused by camera motion.
>
> In scenarios where large motion is caused by camera movement, the impact of object motion on the overall scene change is relatively minor. Moreover, due to the short temporal interval between adjacent frames in video interpolation tasks, we follow the linear motion assumption of previous works.
>
> To evaluate the generalization ability of our method across different scenarios, we provide video interpolation results on various scenes via an anonymous link in the appendix. The experimental results demonstrate that our method is robust across a wide range of scenarios.
>
> ---
>
> ## [W4 & Q2]  VFIMamba
>
> We evaluate the performance of VFIManba on a large motion benchmark. The results show that under extreme scenarios, its performance is inferior to that of our method.
>
> | Method     | X-text-L 2K     | X-text-L 4K     | SNU-FILM-L hard | SNU-FILM-L extreme | Xiph-L 2K       | Xiph-L 4K       |
> |------------|------------------|------------------|------------------|---------------------|------------------|------------------|
> | Manba-VFI  | 31.58 / 0.9169  | 30.50 / 0.9077  | 28.80 / 0.9208  | 23.41 / 0.8300     | 30.72 / 0.8780  | 28.62 / 0.8111  |
>
> ---
>
> ## [W5 & Q1] Limited Visualization
>
> Thanks to the reviewers' suggestions, we presented the result videos on the anonymous website (https://anonymous.4open.science/w/1018-B854/) in the appendix section of the main paper.
>
> We sincerely apologize for the misunderstanding caused by our insufficient explanation of the method pipeline. To clarify our approach, we provide a more detailed and clearer description of the overall process below:
>
> **(I)** We utilize VGGT to capture the static background alignment between different frames. Based on this alignment, we linearly estimate the intermediate point cloud $\mathcal{P}^t$.
>
> **(II)** Using the QGS head, we generate $\mathcal{G}^t$, and then compute the normal vector and curvature attributes of each point based on Equation ($\textcolor{red}{4}$), leading the QGS that incorporates detailed geometric information.
>
> To further improve the accuracy of frame alignment, we introduce normal vectors as geometric constraints, a practice that is widely adopted in traditional 3D reconstruction methods.
>
> However, in regions with significant geometric variation, such as object boundaries or areas with large motion, normal-based constraints alone may be insufficient, potentially introducing errors and causing over-smoothing.
>
> To mitigate this issue, we introduce a curvature map to identify regions with highlighted geometric changes (i.e., areas where frames are not well aligned). This allows us to provide stronger supervision in these regions, guiding the model to achieve more accurate alignment and reconstruction.
>
> **(III)** By rendering the color, normal vector, and curvature attributes of the QGS onto the camera plane through Equations ($\textcolor{red}{7}$) and ($\textcolor{red}{8}$), we obtain the corresponding color map, normal map, and curvature map.
>
> **Intuitive Explanation**
> As shown in the ablation study (Fig. 8) in the main paper, the curvature maps highlight regions with significant geometric changes, demonstrating that our method effectively focuses on areas with meaningful structural variations. This supports the intuition that curvature serves as a strong indicator for geometric alignment and detail preservation.
>
> In addition, we provide a visualization of the projection process from QGS to the image plane. These visualizations are available through the anonymous website linked in the appendix.
>
> *Thank you again for your time and effort in reviewing our work!*

---

> > ### Comment · Reviewer_HLES · 2025-08-05
> >
> > I thank the author for patiently addressing my questions, and I have decided to improve my score.

---

### Official Review · Reviewer_eXnA · 2025-07-03

**Clarity:** 3
**Significance:** 2
**Originality:** 2
**Rating:** 4
**Confidence:** 3

**Summary:**

This work presents a new frame interpolation model based on modeling a generalized notion of pixel, named Quadratic Gaussians (QG). Specifically, the model employs foundation 3D reconstruction model (e.g.., DUSt3R) to estimate the 3D coordinates (or Gaussian means) and the camera poses of given video frames, then additionally employ a head network to predict Gaussian splatting parameters (such as the opacity, covariances, etc.) based on the internal features and the outputs of the 3D reconstruction model. Finally, the surface properties such as normals and curvatures are derived in a rule-based manner following their mathematical definition (as given in Equation 4). These features, along with the predicted Gaussian splatting parameters, are rasterized to 2D planes to render the final image (i.e., the interpolation target frame).

The experiments are performed in several standard VFI benchmarks, and the proposed method achieves comparable or slightly enhanced performance with respect to recent baseline models.

**Questions:**

- As noted previously, DUSt3R is known not to generalize well on videos capturing dynamic scenes. Is it possible to employ the dynamics-aware variants, such as the recent CUT3R (Wang et al., 2025)? What would be the impact on the performance on the model, if such variants are employed instead of DUSt3R?

- In the current design, the camera pose of the interpolation target is estimated via linearly interpolating the camera poses predicted by the 3D foundation model. Is there any failure case due to noisy camera pose estimation results?

**Ethical Concerns:**

["NO or VERY MINOR ethics concerns only"]

**Final Justification:**

I appreciate the authors' comprehensive response to my review. Their answers successfully addressed my questions, and after reviewing the supplementary material, I find the paper's findings more convincing. As the authors themselves noted, more robust baselines for 3D geometry estimations have become available. I encourage the authors to explore unifying their method with these latest baselines in the final manuscript.

In summary, I am updating my score to Borderline accept. I believe this method is valuable because it proposes a novel approach for video frame interpolation (VFI), exploring the use of Gaussian splats to replace conventional pixels.

**Limitations:**

The proposed design is adapt to the 2 source frame - 1 target frame design for VFI. However, recent foundation 3D reconstruction models (which the proposed method heavily relies on) mainly perform multi-frame inferences to obtain more robust results when affected by complex dynamics. It may not be trivial to generalize the proposed method for these foundation 3D models.

**Quality:**

2

**Strengths And Weaknesses:**

- Strength of the paper includes clear presentation and straightforward model design

- The method directly relies on the outputs of foundation 3D reconstruction model (e.g., DUSt3R), which provides camera parameters and initial 3D coordinates of the pixel points. However, these foundation models are known to be underperforming in videos with dynamic objects, which can potentially harm the generalization performance of the proposed method in various real-world videos

-  The training objective includes reconstructing the surface normal and curvature, however, the details for obtaining ground truth for these features are missing in the current manuscript

---

> ### Author Rebuttal · Authors · 2025-07-30
>
> Thank you for your insightful inquiries, and we will provide detailed responses to each of them below:
>
> ---
>
> ## [W1 & Q1] Selection of  3D reconstruction model
>
> Frame correspondences can be categorized into two types: **static correspondences**, which are primarily caused by background changes due to camera motion, and **dynamic correspondences**, which result from object motion. Existing approaches, such as optical flow and self-attention mechanisms, have certain limitations in modeling scene-level static correspondences.
> They often struggle to capture globally consistent background alignment, which in turn affects the quality of frame interpolation.
> Our method mainly focuses on addressing large-scale background displacements induced by camera motion.
>
> In the appendix, we demonstrate the frame alignment capabilities of different 3D reconstruction models through an anonymous project website (link: https://anonymous.4open.science/w/1018-B854/) in the appendix section of the main paper. Specifically, **VGGT** shows significantly better performance than **CUT3R** in modeling **static correspondences**, exhibiting stronger background alignment. However, for dynamic objects, both VGGT and CUT3R suffer from a certain degree of object overlap.
>
> Due to the alignment of static correspondences and the short temporal interval between adjacent frames in video interpolation tasks, we follow the linear motion $\mathcal{P}^t= (1-t)\mathcal{P}^0 + t \mathcal{P}^1$ assumption of previous works.  In addition, the ablation study in the main paper (Table 3) further confirms the performance advantage of VGGT over CUT3R, where VGGT achieves a PSNR of 24.2 compared to 24.11 for CUT3R. Based on these observations, we adopt VGGT as our 3D reconstruction model.
>
> ---
>
> ## [W2] Loss Function
>
> We are sorry for the confusion caused by the insufficient loss function description. The loss function primarily consists of the RGB reconstruction loss, which is supervised by **the ground truth frame** $I^t_{gt}$.
> To further regularize the geometric alignment, we incorporate a curvature-aware normal term $\mathcal{L}_{kn}$.
>
> The $\mathcal{L}_{kn}$ is designed to enforce geometric alignment in 3D space. However, in regions with significant geometric variation, such as object boundaries or areas with large motion, normal-based constraints alone may be insufficient, potentially introducing errors and causing over-smoothing.
>
> To mitigate this issue, we introduce a curvature map to identify regions with highlighted geometric changes (i.e., areas where frames are not well aligned). This allows us to provide stronger supervision in these regions, guiding the model to achieve more accurate alignment and reconstruction. In addition, Figure 8 in the main paper provides further evidence that the curvature map successfully captures regions of geometric misalignment.
>
> The computation formulas for surface normals and curvature are provided in Equation ($\textcolor{red}{4}$) of the main paper.
> In addition, Figure  $\textcolor{red}{8}$ in the main paper provides further evidence that the curvature map successfully captures regions of geometric misalignment.
>
> ---
>
> ## [Q2] Camera pose
>
> We thank the reviewer for this insightful question and pay attention to our work.
>
> Our method performs video frame interpolation using only two input frames. As a result, the estimated camera poses are relatively sparse, and common issues related to pose ordering in multi-frame sequences do not arise.
> In current video interpolation experiments, we have not observed significant performance degradation caused by camera pose noise.
> Moreover, recent works [1, 2, 3] demonstrate that sparse and unposed image pairs can still yield high-quality results in 3D reconstruction and novel view synthesis.
>
> [1] Zhang, Shangzhan, et al. "Flare: Feed-forward geometry, appearance and camera estimation from uncalibrated sparse views." Proceedings of the Computer Vision and Pattern Recognition Conference. 2025.
>
> [2] Ye, Botao, et al. "No pose, no problem: Surprisingly simple 3d gaussian splats from sparse unposed images." ICLR 2025.
>
> [3] Smart, Brandon, et al. "Splatt3r: Zero-shot gaussian splatting from uncalibrated image pairs." Proceedings of the Computer Vision and Pattern Recognition Conference. 2025.
>
> ---
>
> *We hope our clarifications address your concerns and demonstrate the improved quality of our paper! Thank you again for your valuable time!*

---

> > ### Comment · Reviewer_eXnA · 2025-08-05
> >
> > I appreciate the authors' comprehensive response to my review. Their answers successfully addressed my questions, and after reviewing the supplementary material, I find the paper's findings more convincing. As the authors themselves noted, more robust baselines for 3D geometry estimations have become available. I encourage the authors to explore unifying their method with these latest baselines in the final manuscript.
> >
> > In summary, I am updating my score to Borderline accept. I believe this method is valuable because it proposes a novel approach for video frame interpolation (VFI), exploring the use of Gaussian splats to replace conventional pixels.

---

### Official Review · Reviewer_csF4 · 2025-07-03

**Clarity:** 3
**Significance:** 2
**Originality:** 3
**Rating:** 6
**Confidence:** 4

**Summary:**

The paper introduces a novel approach to video frame interpolation by representing video frames as 3D differential surfaces. The authors introduce a new framework, taking inspiration by Abena et al. [22] and Zhang et al. [23], to map frames to Quadratic Guassian responsible for global surface-level alignment unlike previous methods for better object correspondence alignment. Surface properties are rasterized onto the interpolated camera plane and a custom loss function enforces proper alignment therein. Several experiments underline the effectiveness of the proposed method both quantitavely and qualitatively.

**Questions:**

1. How was the $\alpha$ loss weighting factor chosen? What would happen if it was set to a very large value or a very small value? how would results change?
2. Why have you decided to train on two datasets and compare with methods that are only trained on one dataset? Could you elaborate on this and provide more insights?
3. Out of curiosity, how well does your model generalize to larger resolutions or how much retraining it would require (interesting as far as FILM is able to generalize quite well for larger resolutions and only requires very little continued training on larger resolutions to hand frame interpolation fairly well give it is only trained on small resolutions)?

**Ethical Concerns:**

["NO or VERY MINOR ethics concerns only"]

**Final Justification:**

I appreciate the author's feedback and active engagement in the discussion with both myself and the other reviewers. I now believe the submission is overall quite strong. The additional evaluations were insightful, and the responses provided were clear and satisfactory. I am therefore raising my rating.

**Limitations:**

I highly appreciate the effort and openness for doing a limitations section and mentioning weaknesses. However, this could be more specific and explicit with limiting examples rather than keeping it fairly general (or at least a rewording would make limitations more clear in my opinion). For example: an inherent limitation of the method is the ability to infer geometry from appearance which is not possible in every case, especially in very fine detailed repetitive patterns in rather small areas.

**Paper Formatting Concerns:**

No concerns.

**Quality:**

2

**Strengths And Weaknesses:**

Strengths:
1. It is the first paper to explore differential geometry in large motion video frame interpolation by treating frames as 3D surfaces.
2. The architecture itself is modular making it easier to extend and change the method in follow-up work making it more accessible to the entire audience.
3. Experimental results support the reason for considering differential geometry. The presented results highlight cases where the proposed method outperforms previous SOTA models.

Weaknesses:
1. Diffusion models, specifically DDPM, are mentioned in the beginning of the paper, however, they are not used in comparison. Possibly due to being time consuming and being challenging for real-time inference (I assume so by reading your Related work section). Yes, in general they are not real-time worthy but as your method fails to be real-time as well it is not common practice to remove good performing diffusion models from comparison because of that exact reason. "Video Interpolation with Diffusion Models" by Jain et al. is a paper I would like to highlight that is also able to perform very crisp interpolation of two frames.
2. In Table 1 you compare methods that have not been trained on the same dataset. It would however be very interesting to see how your model would perform if it was also only trained on the Vimeo-90K dataset in comparison and mention that as training on two datasets theoretically allows for a wider variety of data that the other methods did not have access to. I think it would be fair to neglect the methods SGM-VFI and VFIMamba-S in that case as I see only EMA-VFI-S to be your closer competitor ranking second or first more often than the other two methods combined (It is still fine to keep it in the supplementary material). This is the main reason for the score but I am happy to adjust it if relevant comparisons can be shown.
3. Only 2 visual comparisons are made which make it hard to judge. Of course in these the model outperforms the others but it would be nice to see a more extensive comparison.
4. The results on the anonymized GitHub repository and the paper itself do not match. Either there was some compression affecting the result or there is a bigger problem as the shirt of the skater for example is very well readable in the paper images but in the individual images in the gif on the website they are super blurry and not readable at all.
5. This is a weakness that applies to any frame interpolation method more or less: Usually $t \ \in \ (0,1)$ is set as 0.5 the entire time. However, as you have slow-motion videos or videos with 1000fps at hand would it not be helpful to train the network with varying $t$? The problem that arises from setting $t = 0.5$ the entire time occurs when you want to generate a sequence of images between a starting and an ending image. The quality always relies on the resulting mid-frame generated whereas choosing a free $t \ \in \ (0,1)$ could in theory help supervise smaller movements and preserve more details as well as making the method more robust due to requiring only the starting and ending image for interpolating at any $t$. Did you just follow this way because it is established? (I do understand that this makes it a lot easier when comparing but it makes it harder for the later inference use.)

---

> ### Author Rebuttal · Authors · 2025-07-30
>
> We appreciate your invaluable insights and thoughtful comments. In the following sections, we address the questions you have raised:
>
> ---
>
> ## [W1] Comparison of Diffusion-Based Models
>
> We conduct a comparative study of two diffusion-based video frame interpolation approaches, namely VIDIM and LDMVFI, evaluating their performance on both standard benchmarks and large motion benchmarks.
>
> The results demonstrate that diffusion models exhibit suboptimal performance in dynamic scenarios, especially when significant background motion arises due to camera movement. In addition, current diffusion-based video frame interpolation approaches lack real-time capability, thereby constraining their applicability in real-world deployments.
>
> **Stand Benchmark**
>
> | Method  | SNU-FILM easy | SNU-FILM medium | SNU-FILM hard | SNU-FILM extreme  | Runtimes (s) | FLOPs (T) |
> |---------|----------------------------|------------------------------|----------------------------|-------------------------------|---------------|------------|
> | VIDIM   | 38.103 / 0.976             | 33.280 / 0.961               | 27.870 / 0.900             | 23.001 / 0.830                | 9.28          | 10.8       |
> | LDMVFI  | 38.679 / 0.987             | 33.998 / 0.970               | 28.547 / 0.917             | 23.932 / 0.837                | 8.48          | 9.9        |
>
> **Large Motion Benchmark**
>
> | Method | X-text-L 2K  | X-text-L 4K | SNU-FILM-L hard | SNU-FILM-L extreme | Xiph-L 2K  | Xiph-L 4K |
> |--------|-------------------------|--------------------------|------------------------------|---------------------------------|------------------------|------------------------|
> | VIDIM  | 29.22 / 0.871           | 28.15 / 0.861            | 25.71 / 0.880                | 21.52 / 0.794                   | 27.83 / 0.849          | 27.00 / 0.780          |
> | LDMVFI | 29.71 / 0.875           | 28.72 / 0.867            | 26.17 / 0.888                | 21.66 / 0.794                   | 28.37 / 0.864          | 27.80 / 0.787          |
>
> ---
>
> ## [W2 & Q2]  The same dataset
>
> SGM-VFI introduces a benchmark dataset designed to evaluate video frame interpolation under large motion scenarios. To ensure a fair comparison with SGM-VFI in such challenging settings, we follow its training strategy and jointly train our model on both the V and X datasets.
>
> We appreciate the reviewer’s suggestion. In response, we add evaluation results for a model trained only on the V dataset, tested on both standard benchmarks and the large-motion benchmark.
> Our method demonstrates powerful performance on the large-motion benchmark.
> And it demonstrates competitive performance on the standard data benchmark.
>
> **Stand Benchmark**
>
> |   | Vimeo-90K | UCF101 | SNU-FILM easy | SNU-FILM medium | SNU-FILM hard  | SNU-FILM extreme |
> |--------|------------------------|---------------------|-----------------------------|-------------------------------|-----------------------------|-------------------------------|
> | Ours   | 36.09 / 0.980          | 35.40 / 0.9702      | 39.99 / 0.9907              | 36.11 / 0.9800                | 30.95 / 0.9400              | 26.00 / 0.8681                |
>
> **Large Motion Benchmark**
>
> |   | X-text-L 2K | X-text-L 4K  | SNU-FILM-L hard | SNU-FILM-L extreme | Xiph-L 2K | Xiph-L 4K  |
> |--------|-------------------------|--------------------------|------------------------------|---------------------------------|------------------------|------------------------|
> | Ours   | 31.33 / 0.9012          | 30.12 / 0.9066           | 29.05 / 0.9213               | 24.20 / 0.8400                  | 31.20 / 0.8814         | 29.16 / 0.8197         |
>
> ---
>
>
> ## [W3 & W4] Visual comparisons
>
> We update the lossless GIFs on the anonymous website ( https://anonymous.4open.science/w/1018-B854/) in the appendix section of the main paper.
>
> ---
>
>
> ## [W5] How to do continuous interpolation
>
> This is a good question. We thank the reviewer for raising it.  The continuous interpolation follows the pipeline outlined:
>
> **(I)** Input Frames: We take two input video frames, $I^0 \in \mathbb{R}^{H \times W}$ and $I^1 \in \mathbb{R}^{H \times W \times 3}$, where  H and W denote the height and width of the image, respectively.
>
> **(II)**  Point Cloud: Frame correspondences can be categorized into two types: **static correspondences**, which are primarily caused by background changes due to camera motion, and **dynamic correspondences**, which result from object motion.
> VGGT demonstrates strong performance in modeling **static correspondences**. We leverage this capability to obtain the point clouds $\mathcal{P}^0 \in \mathbb{R}^{HW \times 3}$  and $\mathcal{P}^1 \in \mathbb{R}^{HW \times 3}$  aligned based on static correspondences, where $HW$ is the number of points (pixels), $3$ represents the 3D spatial coordinates.
> In addition, we visualize the scene by constructing $[ \mathcal{P}^0 || \mathcal{P}^1 ]$ to verify the quality of the static correspondence alignment in an anonymous project website( https://anonymous.4open.science/w/1018-B854/). Here, the symbol $||$  denotes the concatenation operation, i.e., torch.cat() in PyTorch.
>
> **(III)** Linear Motion:
> Due to the alignment of static correspondences and the short temporal interval between adjacent frames in video interpolation tasks, we follow the linear motion assumption of previous works. Based on this, we compute the interpolated point cloud at time  $t \in [0, 1]$ as: $\mathcal{P}^t = (1-t) \mathcal{P}^0 + t \mathcal{P}^1$. By continuously varying $t$, we generate a sequence of intermediate point clouds
> $\mathcal{P}^t \in \mathbb{R}^{HW\times3}$.
>
> **(IV)** Rendering Intermediate Frame: The interpolated point cloud $\mathcal{P}^t$  is then fed into the QGS head, which renders the intermediate frame $I^t$. We compute the loss between the rendered frame $I^t$  and the ground truth intermediate frame $I^t_{gt}$.
>
> To intuitively understand the interpolation process, we visualize the entire pipeline. The visualization results are provided in the Appendix (https://anonymous.4open.science/w/1018-B854/).
>
> ---
>
>
> ## [Q1] How was the loss weighting factor chosen?
>
> Our loss function primarily consists of the RGB reconstruction loss, which is supervised by the ground truth image $I_{gt}^{t}$.
> To further regularize the geometric alignment, we incorporate a curvature-aware normal term $\mathcal{L}_{kn}$. Consistent with the settings adopted in previous methods, such as 2D GS and QGS, we set the weight of the curvature-aware normal term to 0.2.
> An excessively large weight for the curvature-aware normal term makes the overall loss function difficult to optimize, while a weight that is too small may leading poor geometric alignment.
>
> The curvature-aware normal term $\mathcal{L}_{kn}$ is designed to enforce geometric alignment in 3D space. However, in regions with significant geometric variation, such as object boundaries or areas with large motion, normal-based constraints alone may be insufficient, potentially introducing errors and causing over-smoothing.
>
> To mitigate this issue,  we introduce a curvature map to identify regions with highlighted geometric changes (i.e., areas where frames are not well aligned). This allows us to provide stronger supervision in these regions, guiding the model to achieve more accurate alignment and reconstruction.  In addition, Figure $\textcolor{red}{8}$ in the main paper provides further evidence that the curvature map successfully captures regions of geometric misalignment.
>
> ---
>
> ## [Q3] Out of curiosity, how well does your model generalize to larger resolutions?
>
> We thank the reviewer for this curious question and pay attention to our work.
> We evaluated the performance of our method on high-resolution datasets. In Table 2 of the main paper, the X-Test (Xiph) dataset includes video sequences with 2K and 4K resolutions, which are used to assess the effectiveness of our method in high-resolution scenarios.
>
> Specifically, we adopt a patch-based inference strategy, where overlapping patches are split from high-resolution frames. This approach allows our model to handle large resolution inputs efficiently without exceeding memory constraints. This supports the feasibility and scalability of our approach.
>
> For 4K-specific research, we recommend Zhang et al. [1], which proposes an efficient method for high-resolution panorama synthesis.
>
> [1] Zhang, Cheng, et al. "Pansplat: 4k panorama synthesis with feed-forward gaussian splatting." Proceedings of the Computer Vision and Pattern Recognition Conference. 2025.
>
> ---
>
> *Hope our explanation and experiments are able to address your inquiries. Please don't hesitate to reply if you have any further concerns. We will integrate all your valuable suggestions into our revision! Thank you!*

---

> > ### Comment · Reviewer_csF4 · 2025-08-05
> >
> > I appreciate the authors' time and effort in conducting additional evaluations and thoughtfully addressing my questions. The new comparisons with VIDIM and LDMVFI provide valuable insights. It's particularly impressive that the model trained solely on Vimeo still achieves strong performance, highlighting its robust generalization capabilities and further strengthening the paper. Additionally, the clarification of interpolation for $t \in (0, 1)$, along with the supplementary visualizations, is clearly presented and helpful. Overall, I would be happy to raise my rating.
> >
> > Would it also be possible to include LPIPS scores for your method? Based on the visual results, I would expect them to reflect favorably on your approach and believe they could further strengthen the paper.

---

> ### Author Response · Authors · 2025-08-05
>
> We sincerely thank the reviewer’s feedback and are more than happy to include the LPIPS metric in our evaluation.
>
>
> **standard benchmark**
>
> | Method | Vimeo-90K | UCF101 | SNU-FILM easy | SNU-FILM medium | SNU-FILM hard | SNU-FILM extreme |
> |--------|-----------|--------|----------------|------------------|----------------|-------------------|
> | Ours   | 0.0196    | 0.032  | 0.0191         | 0.0334           | 0.06236        | 0.12026           |
> | LDMVFI | 0.0173    | 0.026  | 0.014          | 0.028            | 0.060          | 0.123             |
>
>
>
> **large motion benchmark**
>
> | Method | X-text-L 2K | X-text-L 4K | SNU-FILM-L hard | SNU-FILM-L extreme | Xiph-L 2K | Xiph-L 4K |
> |--------|-------------|-------------|------------------|---------------------|-----------|-----------|
> | Ours   | 0.64501     | 0.6567      | 0.0816           | 0.15964             | 0.1545    | 0.1756    |
> | LDMVFI | 0.6626      | 0.6701      | 0.0823           | 0.15933             | 0.1580    | 0.1785    |
>
> As the results indicate, our method demonstrates strong competitiveness against LDMVFI, a diffusion-based model. It is well established that diffusion models, by their generative nature, often excel at producing high-frequency visual details, thus gaining an inherent advantage in perceptual metrics like LPIPS.
>
> However, our approach achieves highly comparable, and in some large-motion cases, superior LPIPS scores. This is primarily attributed to the curvature-aware loss term. While traditional methods relying on L1/L2 loss tend to produce overly smoothed results that penalize LPIPS scores, the curvature-aware loss term enhances the modeling of geometric structures and fine-grained textures.
>
> *Once again, we thank the reviewer for the kind feedback.*

---

### Note · Authors · 2025-08-12

Dear Area Chair and Reviewers,

We sincerely thank you for your valuable time and insightful feedback on our manuscript. Below is a **summary** of the main concerns and our responses.

----

## csF4

**Concerns**

* The model is trained on two datasets, whereas some baselines use only one.

* A comparison with diffusion-based models is absent.


**Rebuttal**

* We provide new experimental results trained on the Vimeo-90K, which demonstrate strong performance.

* We added new comparisons with two diffusion models.

**Final**

The reviewer is satisfied and expresses a willingness to raise rating.

---
## eXnA

**Concerns**

* The method relies on a 3D reconstruction model (VGGT) that may not be robust in dynamic scenes.

* The source of ground truth for the proposed surface normal and curvature loss was unclear.

**Rebuttal**

* We clarify that our focus is on static background motion, where the chosen model (VGGT) excels, and support this with ablation studies.

* We explain that the geometric loss functions as a regularizer, with normal and curvature values being analytically computed from the predicted surface itself, rather than being supervised by an external ground truth.

**Final**

The reviewer raises the score to "Borderline accept."

---
## HLES

**Concerns**

* The theoretical background is difficult to follow.

*  Lacks a comparison with VFIMamba on the large motion benchmark.

* The theoretical basis assumes rigid motion, raising questions about real-world applicability.

**Rebuttal**

* We provide a detailed explanation of the background.

* We conduct new experiments comparing our method against VFIMamba, showing superior performance on the large motion benchmark.

* We clarify our assumptions regarding motion and provide additional results to demonstrate generalization.

**Final**

The reviewer decides to improve the score.

---
## aERN

**Concerns**

* The paper lacks comparisons to recent related works that use Gaussian Splatting (e.g., Splatt3r, No pose).

* Enhancing Paper Readability: A clearer connection between the theoretical section and the methodology is recommended.

**Rebuttal**

* We provide new comparisons against both Splatt3r and Nopose.

* We agree to the reviewer's suggestion to enhance clarity.

**Final**

The reviewer is satisfied with the experiments and commitment to improving the paper's clarity.

----

Thank you to the reviewers and Area Chair for your diligent work and invaluable feedback.

Sincerely,

The Authors

---

### Decision · Program_Chairs · 2025-09-17

**Decision:**

Accept (poster)

**Comment:**

The paper received 4 reviews. The authors-reviewers discussion clarified all points raised in the initial reviews. A consensus that the paper has significance and will be ready for publication pending minor changes was reached and the scores were revised accordingly. The paper can be accepted as poster.